# High-throughput prediction of protein conformational distributions with subsampled AlphaFold2

Gabriel Monteiro da Silva [1], Jennifer Y. Cui [1], David C. Dalgarno [2], George P. Lisi [1,3] & Brenda M. Rubenstein [1,3] ✉

This paper presents an innovative approach for predicting the relative populations of protein conformations using AlphaFold 2, an AI-powered method that has revolutionized biology by enabling the accurate prediction of protein structures. While AlphaFold 2 has shown exceptional accuracy and speed, it is designed to predict proteins' ground state conformations and is limited in its ability to predict conformational landscapes. Here, we demonstrate how AlphaFold 2 can directly predict the relative populations of different protein conformations by subsampling multiple sequence alignments. We tested our method against nuclear magnetic resonance experiments on two proteins with drastically different amounts of available sequence data, Abl1 kinase and the granulocyte-macrophage colony-stimulating factor, and predicted changes in their relative state populations with more than 80% accuracy. Our subsampling approach worked best when used to qualitatively predict the effects of mutations or evolution on the conformational landscape and well-populated states of proteins. It thus offers a fast and cost-effective way to predict the relative populations of protein conformations at even single-point mutation resolution, making it a useful tool for pharmacology, analysis of experimental results, and predicting evolution.

Proteins are essential biomolecules that carry out a wide range of functions in living organisms. Understanding their three-dimensional structures is critical for elucidating their functions and designing drugs that target them[1]. Historically, experimental techniques such as X-ray crystallography, nuclear magnetic resonance (NMR) spectroscopy, and electron microscopy have been used to determine protein structures[2–4]. However, these methods can be time-consuming, technically challenging, and expensive, and may not work for all proteins[5]. To meet this challenge, ab initio structure prediction methods, which use computational algorithms to predict protein structures from their amino acid sequences, have been developed[6]. For many years, ab initio structure prediction methods have relied on physics-based algorithms to predict stable protein structures[7]. Although successful, these methods are challenged by larger and more complex proteins[8].

The recent development of machine learning algorithms has significantly improved the speed of protein structure prediction[9,10]. One of the most remarkable achievements in this area is the AlphaFold 2 (AF2) engine developed by DeepMind, which uses a deep neural network to predict ground state protein structures from amino acid sequences[11,12]. AlphaFold 2 was trained using large amounts of experimental data and incorporates co-evolutionary information from massive metagenomic databases[11]. Its accuracy has revolutionized the field of protein structure prediction[11,13,14], opening up new possibilities for drug discovery and basic research with clear consequences for human health[15,16].

However, a series of studies have found that the default AF2 algorithm is limited in its capacity to predict alternative protein conformations and the effects of sequence variants[17,18]. Although AF2's

[1]Brown University Department of Molecular and Cell Biology and Biochemistry, Providence, RI, USA. [2]Dalgarno Scientific LLC, Brookline, MA, USA. [3]Brown University Department of Chemistry, Providence, RI, USA. ✉e-mail: brenda_rubenstein@brown.edu

inability to predict multiple conformations is unsurprising given its initial scope, the capacity to make predictions of different conformations would be as revolutionary as the capacity to accurately predict ground states. Phenomena that involve different conformations such as fold-switching and order-disorder transitions are ubiquitous across the proteome[19,20] and are directly tied to the activity of many macromolecules[21]. Moreover, methods that can rapidly predict multiple conformations may have the potential to revolutionize drug discovery by uncovering substantially more drug targets[22]. To fully realize this potential, methods like AF2 will need to account for the relative populations of different conformations (states) since the conformational equilibrium of drug receptors is directly related to their affinities for drugs[23,24]. A prime example of this relationship is Imatinib[25], a tyrosine inhibitor whose exceptional selectivity for Abl1 kinase was found to be caused by the enzyme's significant preference for conformational states that facilitate Imatinib recognition and subsequent induced-fit binding[26].

In an attempt to realize this potential, researchers have devised new ways of employing the AF2 method to detect conformational changes, with significant success in a few test cases[27–31]. Although AF2 cannot conventionally predict conformational ensembles, researchers have found that sub-sampling the input multiple sequence alignments (MSAs) and increasing the number of predictions leads to structural ensembles that capture different physiologically-relevant conformations from the same sequence[27]. These predictions can be used as seeds in molecular dynamics (MD) simulations seeking to explore larger swaths of the conformational space and the relative populations of each predicted state[28]. Despite being a significant improvement over methods that only predict ground states, methods such as these still rely on expensive MD simulations to infer most relative state population information, which comes at a significant cost compared to simply running a prediction engine.

Here, we show that these MD simulation steps may be unnecessary if the goal is to discover major alternative conformations and their relative populations in a high-throughput fashion, such as for contrasting differences in the dynamics of orthologs or allelic series of a protein of interest. We take inspiration for this work from the observation that proteins from the same evolutionary line can have differences in relative state populations that are strongly correlated with the genetic distance between them[26]. Since AF2 works by decoding co-evolutionary signals[15] and previous works have suggested that subsampling MSAs leads to accurate predictions of different conformations of the same protein[27], it seems reasonable to hypothesize that some instructions for conformational sampling could be decoded from sequence data alone. If this hypothesis holds true, AF2 and other AI-based methods could be capable of quantifying sequence-encoded dynamic signals, which would make it possible to predict not only alternative conformations of the same protein, but also changes in its relative state populations.

With this as motivation, we show how subsampling multiple sequence alignments can generate ensembles of protein conformations (see Fig. 1 for an overview of our method) and systematically test AF2's capacity to predict sequence-induced differences in the conformational distributions of the Abl1 tyrosine kinase core and of the granulocyte-macrophage colony-stimulating factor (GMCSF). Diverging from previous works, as a first example, we focus on detecting changes in the active state population across the Src kinase to Abl1 evolutionary line and test our ability to predict the effects of single and double point mutations known or suspected to shift state distributions. Crucially, we found that subsampled AF2 can qualitatively predict both the positive and negative effects of mutations on the active state populations of kinase cores with up to eighty percent accuracy. We also found that AF2 predicts most of the activation loop intermediate states in the active-to-inactive transition of the kinase core with an ensemble that is comparable to that obtained from

enhanced-sampling MD simulations. As a second example, we predicted changes in the conformational ensemble of GMCSF, a protein with minimal known homology, in response to point mutations. Our predictions strongly correlated with experimentally-determined NMR results, further showcasing subsampled AF2's remarkable capacity to decode signals pertaining to conformational changes even when sequence data is scarce. Altogether, these results highlight the strong, yet untapped potential of AF2 for predicting changes in the conformational ensembles of proteins, which will have substantial impacts on the fields of biophysics and drug discovery.

## Results

### Optimizing MSA subsampling to predict kinase core conformational ensembles

In recent years, multiple groups have observed that AF2 with different parameters and MSA depths is capable of predicting conformational changes based on sequence data alone[27,28]. These alternative AF2 pipelines share the principle of subsampling MSAs to modulate co-evolutionary signals at different structural domains[27]. In its standard implementation, AF2 takes as input a target sequence and a corresponding multiple sequence alignment. An arbitrary number of sequences (defined by the *max_seq* parameter) are randomly selected from the master MSA (the target sequence is always selected), and the remaining sequences are clustered around each of the selected sequences using a Hamming distance. Both the cluster centers and a sample from each cluster with a length of *extra_seq* are used by AF2 for inference (see Fig. 2). Previous works have shown that a significant reduction in the values of *max_seq* and *extra_seq* from their default values achieves ensemble prediction for a series of model systems[29].

Motivated by these observations, we started our work by systematically testing the accuracy of different AF2 parameter combinations for predicting the Abl1 kinase core structural ensemble. We chose the Abl1 kinase core (residues 229–515) as our first test case due to this protein's extensively documented dynamics[23]. Abl1 is thought to occupy three major conformations with different populations. In solution, Abl1 primarily exists in an active (ground) state. Infrequently, Abl1 will switch to inactive state 1 (I1), and then to inactive state 2 (I2)[23], which strongly binds to Imatinib (Gleevec)[26]. While the change from the ground to I1 state is subtle, the transition from the I1 to I2 state involves considerable backbone rearrangements: the activation loop detaches from its resting position below the C-helix and folds on itself, a change that shifts the activation loop by over 15 Å from its original position as shown in Fig. 3.

To encourage AF2 to generate a full ensemble of Abl1 conformations, we started by compiling an extensive MSA spanning over 600,000 sequences using the JackHMMR algorithm[32] on the wild-type Abl1 kinase core (residues 229–515). This algorithm builds the MSA by querying sequences from the UniRef90[33], Small BFD[34], and MGnify[35] databases. To increase the statistical power of our results, we then ran 32 predictions with independent seeds for each test, and enabled dropouts during inference to sample from the uncertainty of the models. Dropout rates were the same as those found to improve sampling in other studies (10% for the Evoformer module, and 25% for the structural module)[36]. All other parameters were left in their default settings (3 recycles per prediction, 5 models per seed, a total of 160 predictions per run, 3 independent runs with unique seeds, 480 predictions per test).

As described in Supplementary Table 1, we find that a *max_seq:extra_seq* value of 256:512 leads to the most diverse results in terms of activation loop conformations (see Supplementary Figs. 1 and 2). Importantly, the ensemble of activation loop conformations predicted by AF2 with the above parameter set is distributed across the ground state to I2 state transition in Abl1, with no predictions falling significantly outside the boundaries of known activation loop

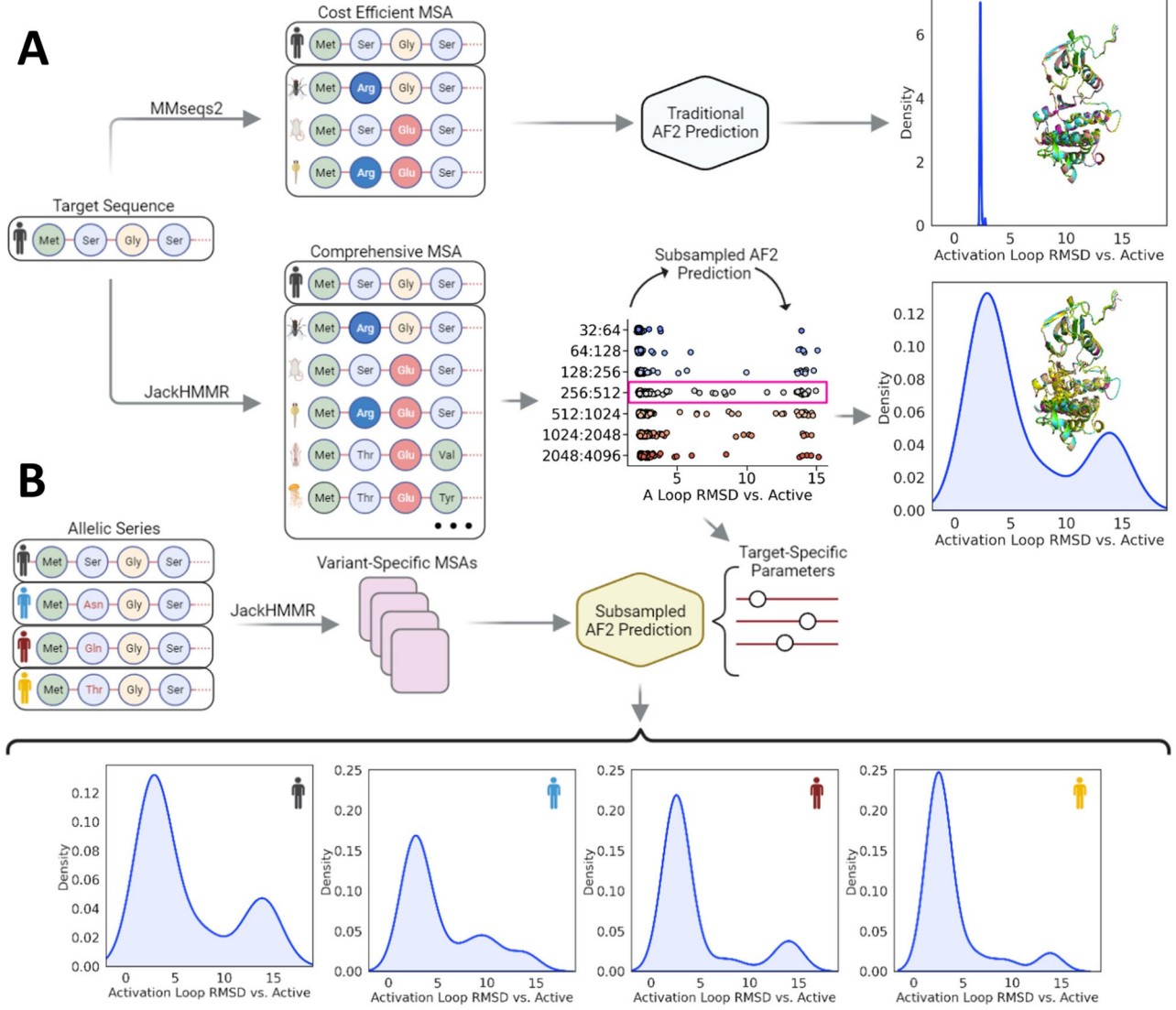

**Fig. 1 | Summary of the subsampled AF2 workflow employed in this study.**
**A** Traditionally, AF2 predicts the structure of a target by using a multiple sequence alignment (MSA). When running AF2 with standard parameters, the predicted structures are often similar to each other even with a large number of independent predictions (seeds). **B** In this study, we show that subsampling deep MSAs causes AF2 to output predictions that occupy different conformations of the same protein, and the predicted frequency of each conformation based upon a range of random seeds strongly correlates with its experimentally-determined relative state population. Figure Created with BioRender.com.

conformations and no blatantly unphysical or misfolded predictions. As a further test of the claim that we are actually predicting conformations along a transition, we compared the ensemble of 160 subsampled AF2 Abl1 predictions to representative snapshots extracted from a I1 to I2 trajectory generated with enhanced-sampling MD simulations of apo Abl1 in solution. Specifics about the methodology used to generate this trajectory are described in Supplementary Figs. 3 and 4 and their accompanying discussion. Representative results from this comparison are illustrated in Fig. 4 and the results of the entire analysis are illustrated in Supplementary Figs. 5–8. Although we expected to observe a range of conformations, the coverage of the activation loop transition is remarkable and suggests the possibility of using AF2 to sample intermediate states and uncover pathways and mechanisms.

### Predicting the conformations of members of a kinase evolutionary line using subsampled AF2
Given our success in predicting the relative population of the Abl1 kinase ground state, we next studied AF2's potential for predicting

conformational distributions without the need for downstream MD simulations. As a basic sanity check, we tested if the Abl1 prediction results were actually the product of AF2 decoding co-evolutionary signal pertaining to relative state populations, or just a fortuitous coincidence. Accordingly, we used the same subsampled AF2 protocol outlined in the Supplementary Materials to predict the relative state populations of the wild-type Src kinase, which is known to occupy the ground (active) state significantly more frequently than Abl1[26] (see Fig. 5), making it an attractive control case. If our hypothesis regarding the potential of subsampled AF2 is indeed correct, we expect that the method will output significantly more predictions of ground state Src than ground state Abl1. Accordingly, we built a large MSA for the Src kinase core (residues 235–497) sequence using the same procedure as described for Abl1 and ran our implementation of subsampled AF2 with it as an input. We then measured the relative population of the Src kinase core ground and I2 states.

Crucially, we found the vast majority of Src kinase core predictions from subsampled AF2 to be in the ground state (for more information about how predictions were binned into different states,

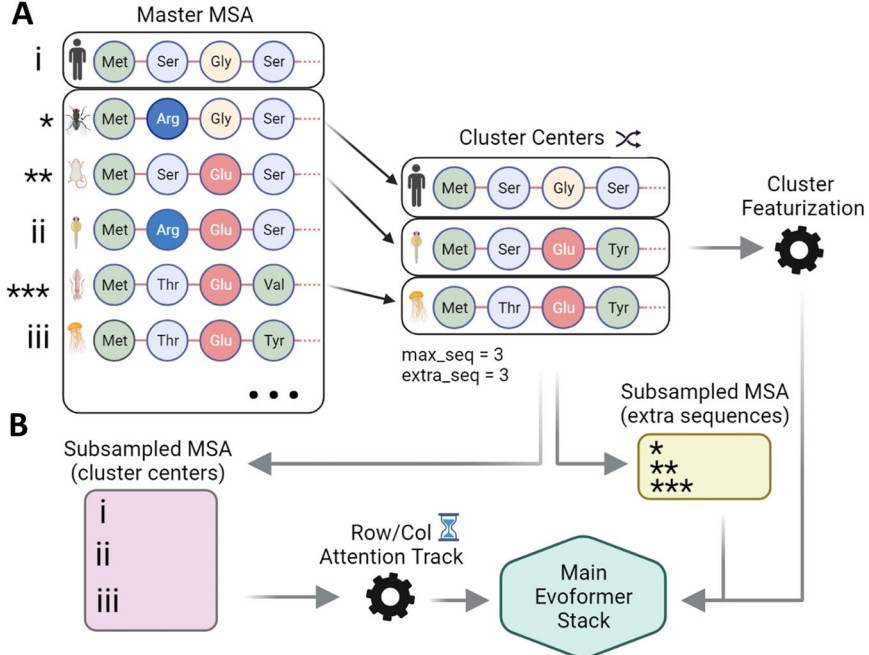

**Fig. 2 | AF2's multiple sequence alignment (MSA) clustering heuristic. A** An MSA of arbitrary length is built from a target sequence and passed to AF2, which randomly selects a number of sequences (defined by *max_seq*) from the input MSA. Each of the selected sequences becomes a cluster center around which the sequences not selected in the previous step are distributed. The target sequence is always selected as a cluster center. The clusters obtained through this process are featurized and relevant statistics are calculated. **B** All of the previously discussed elements are used by AF2 for inference. Cluster features and a number of random non-cluster-center sequences (defined by *extra_seq*) are processed and passed to the Main Evoformer Stack, while the MSA containing the cluster centers is processed, passed to the comparatively expensive row/col attention track, and then finally passed to the Main Evoformer Stack as well. Figure Created with BioRender.com.

see Supplementary Fig. 1), with a predicted relative state population of 97% compared to 89% for Abl1, as summarized in Fig. 6. Interestingly, none of the Src predictions were found to be in the I2 state, although the enzyme is known to infrequently occupy this conformation. This suggests a resolution limitation in using AF2 to predict relative state populations: conformations with very low occupancy such as I2 in Src might be missed by the algorithm in its current implementation. A potential cause of this limitation is the fact that AF2's prediction models were trained on all structures deposited in the Protein Data Bank (PDB) up until 2018, and most of the structures of Src and its orthologs within AF2's training set are in the ground state[37]. We anticipate that fine-tuning or retraining AF2 and similar AI methods with significantly more diverse structural datasets representing different conformational states of a given protein domain could be a viable strategy for increasing the resolution and accuracy of predicting the conformational plasticity of that domain. Additionally, using deeper MSAs in the training could also improve prediction accuracy, in line with recent results that used deep MSAs to achieve higher prediction accuracies than earlier methods[38].

Despite resolution caveats, subsampled AF2 correctly predicted the difference in conformational distributions between the Abl1 and Src kinase cores, lending credence to its promise as a high-throughput method for predicting relative state populations. To shed further light on AF2's capabilities, we applied subsampled AF2 to make predictions for the Anc-AS kinase core (residues 1–263), an Abl1 ancestor with a known conformational state distribution and dynamics[26], and compared the results to the Abl1 and Src cases. The sequences of Abl1, Anc-AS, and Src used to generate the MSAs and as target sequences for AF2 are summarized in Supplementary Fig. 9. While there are other Abl1 ancestors that could be used in this test, there are experimental results for Anc-AS, including a deposited structure in the Protein Databank (4UEU)[26], justifying its choice. For the subsampled AF2 predictions to

be considered accurate, the relative population of the ground state in the Anc-AS predictions should be in between the populations of the same state for the Abl1 and Src predictions, as observed in experimental results. Once again, subsampled AF2 correctly replicated experiments (see Fig. 6), as Anc-AS was predicted to be in the ground state 93% of the time, in-between the frequencies predicted for Src (97%) and Abl1 (89%).

## Predicting the conformations of a kinase allelic series with AF2

While the results we obtained for the Abl1 to Src evolutionary line are promising, an even more impactful application would be predicting how state populations change across an allelic series. This is because many point mutations in proteins are thought to lead to different phenotypes - such as drug resistance - by changing conformational landscapes and relative state populations.

To measure the capacity of subsampled AF2 to fill this niche, we repeated our predictions using a series of Abl1 single and double mutants with well-characterized and significant effects on the relative populations of the ground and I2 states, and contrasted the results with those obtained from the wild-type prediction. Specifically, we tested the method on four mutations that are expected to decrease the population of the ground state (M290L, L301I, F382V, M290L + L301I), and four mutations that are expected to increase it (E255V, T315I, F382L, E255V + T315I). The mutations tested, their locations in Abl1, and their expected effects on ground state populations are summarized in Fig. 5 (see Supplementary Table 2 for more details regarding the expected outcomes of the mutations in the relative state populations of the Abl1 kinase core). The length of the kinase core sequence used as input for this test (229–515) differs slightly from the previous one (235–497) so as to match the length of the constructs tested in the literature. This difference caused a slight variation in the wild-type Abl1 ground state predictions (84% vs. 89%).

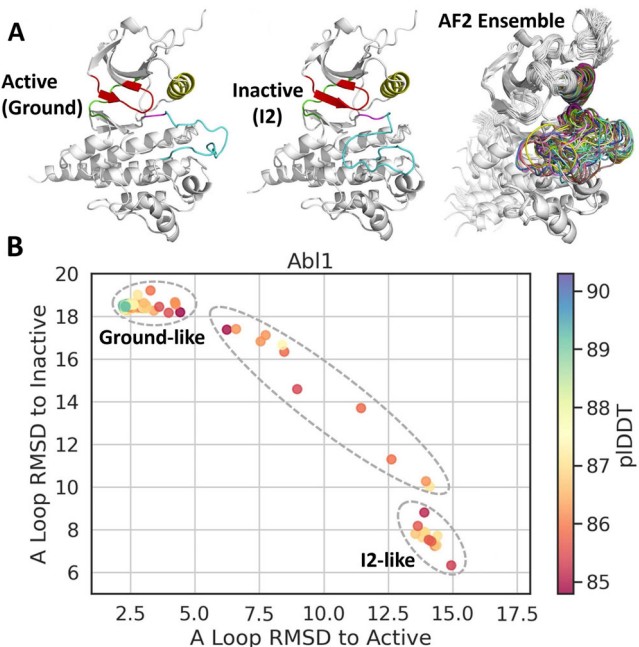

**Fig. 3 | Summary of Abl1 kinase core ensemble prediction results using subsampled AlphaFold2. A** Top: Models of the Abl1 kinase core in its active and inactive conformations. Relevant structural elements are represented in different colors. Cyan: activation loop; red: phosphate-binding loop; yellow: C-helix; green: hinge; and pink: DFG motif. Models are taken from the PDB (6XR6 for Ground, 6RXG for I2)[23]. Top Right: Ensemble of 160 models of the Abl1 kinase core as predicted by subsampled AF2, with the activation loop highlighted in different colors for each prediction. **B** Projection of 160 Abl1 kinase core predictions from subsampled AF2 plotted according to their backbone RMSDs relative to the Abl1 kinase core active (6XR6) and inactive states (6XRG). Data points are colored according to their average predicted local distance difference test score (pLDDT) as calculated by AF2, which is a metric of the confidence of AF2 predictions[11]. Source data are provided as a Source Data file.

AF2 predictions for this allelic series are summarized in Fig. 6. Strikingly, subsampled AF2 correctly predicted a change in the relative state population and its direction in over 80% of the tested cases. Although promising, these results are not without significant caveats. First, the M290L predictions are inaccurate. Specifically, the effects of the mutation on the ground state population are predicted to be the opposite of those seen in experiments. Second, the prediction accuracy only applies in a qualitative sense, as double mutations that are known to significantly increase the ground state population such as M290L + L301I are predicted by subsampled AF2 to increase it only slightly more than single mutations such as M301I, which is known to cause a more subtle increase. We believe that this inaccuracy is a direct consequence of the incorrect M290L prediction, which should also result in the underestimation of the effects of double mutations including M290L. Furthermore, the statistical significance of the results is reduced for the benchmarks in which the mutations are known to reduce ground-state populations. We hypothesize that AF2 performed better at predicting decreases in the ground state population because wild-type Abl1 occupies this state with near 90% occupancy, thus mutations that decrease it generally have a greater magnitude effect on relative state populations than mutations that further increase it.

Finally, considering that the version of AF2 used in this manuscript ships with five different models (see Supplementary Table 3), we asked how well these different models performed at estimating changes in the relative state populations of Abl1 states caused by different point mutations. For a discussion of how different models fared at this challenge, please see Supplementary Fig. 10.

## Predicting GMCSF dynamics with AF2

Considering the success of our Abl1 predictions, we sought to test if the accuracy of these predictions was contingent on the wealth of kinase sequence data, or if we could obtain similar results with significantly less sequence data. To do so, we repeated our prediction workflow but used the sequence of the human granulocyte-macrophage colony-stimulating factor (GMCSF). GMCSF is a 14 kDa monomeric glycoprotein that plays a central role in innate immunity, stimulating a variety of cells in response to pathogens[39]. In contrast to that for kinases, literature regarding GMCSF's structure and dynamics is sparse, and sequence data for homologs or orthologs is orders of magnitude less abundant. Crucially, the MSA built for the wild-type human GMCSF sequence is only 112 sequences long, while that for human Abl1 kinase is over 600,000 sequences using the same parameters and methods (see Fig. 7). This stark contrast represents a useful opportunity to study how the accuracy of subsampled AF2 is modulated by the availability of sequence data.

Previous NMR results have shown that the dynamics of GMCSF's N-terminal helix A are involved in the binding of heparin and other charged molecules[40]. It is thought that the position of helix A with respect to the center of the protein is flexible, usually being tightly packed against helix C through π-π interactions among histidines 15, 83, and 87. Experimental results suggest that this closed configuration is the most stable GMCSF conformation when packed as a crystal[41]. For clarity, this closed conformation will be henceforth referred to as the ground state. It is thought that the π-π interactions are eventually weakened either through intrinsic breathing motions and resultant interactions with the solvent or via induction by other molecules, and helix A is thought to move away from the core. This opening motion exposes part of the GMCSF core to solvent and creates a groove that is thought to be the binding site of heparin and other immune system modulators[40]. This binding-competent GMCSF state will be referred to as the open conformation. Importantly, GMCSF binds heparin much more strongly in an acidic environment, suggesting that helix A dynamics are protonation-dependent[40].

Further NMR experiments (see Supplementary Figs. 11 and 12) have shown that point mutations in the aforementioned histidine triad lead to significant chemical shift perturbations, similar to those caused by reductions in pH[40], hinting at increased occupancy of the open state or other unknown conformations. Importantly, the amplitudes of the changes to the backbone dynamics caused by these mutations vary significantly depending on which histidine of the triad is mutated. Mutating H15 or H83 leads to more pronounced changes in chemical shifts (see Fig. 8). This is expected as H15 and H83 are significantly more buried than H87, meaning that mutations at these positions are likely to be harder for the GMCSF backbone to accommodate. Although the magnitude and distribution of the $^1$H-$^{15}$N peaks observed in Fig. 8 suggest that significant conformational changes involving backbone atoms may occur in GMCSF mutants (especially those involving H15 and H83), we must not discard the possibility that the side chains surrounding the measured atoms might be inducing or involved in the conformational changes. To distinguish between contributions from the main and side chains, further studies exploring GMCSF dynamics with methods that show ensemble averaging of backbone structures such as residual dipolar coupling might be necessary. Finally, mutations H83Y and H83R lead to larger-scale conformational changes (inferred from broadened peaks) than any other mutation in the test set, and considerably more than the H83N mutation, suggesting that specific substitutions at each position induce more significant changes in the GMCSF backbone dynamics (see Supplementary Fig. 11 for a detailed analysis of all mutant CSPs and broadened peaks). These mutations are therefore useful for benchmarking as they represent a tiered challenge of predicting which specific amino acid substitution will lead to the largest changes in GMCSF structure.

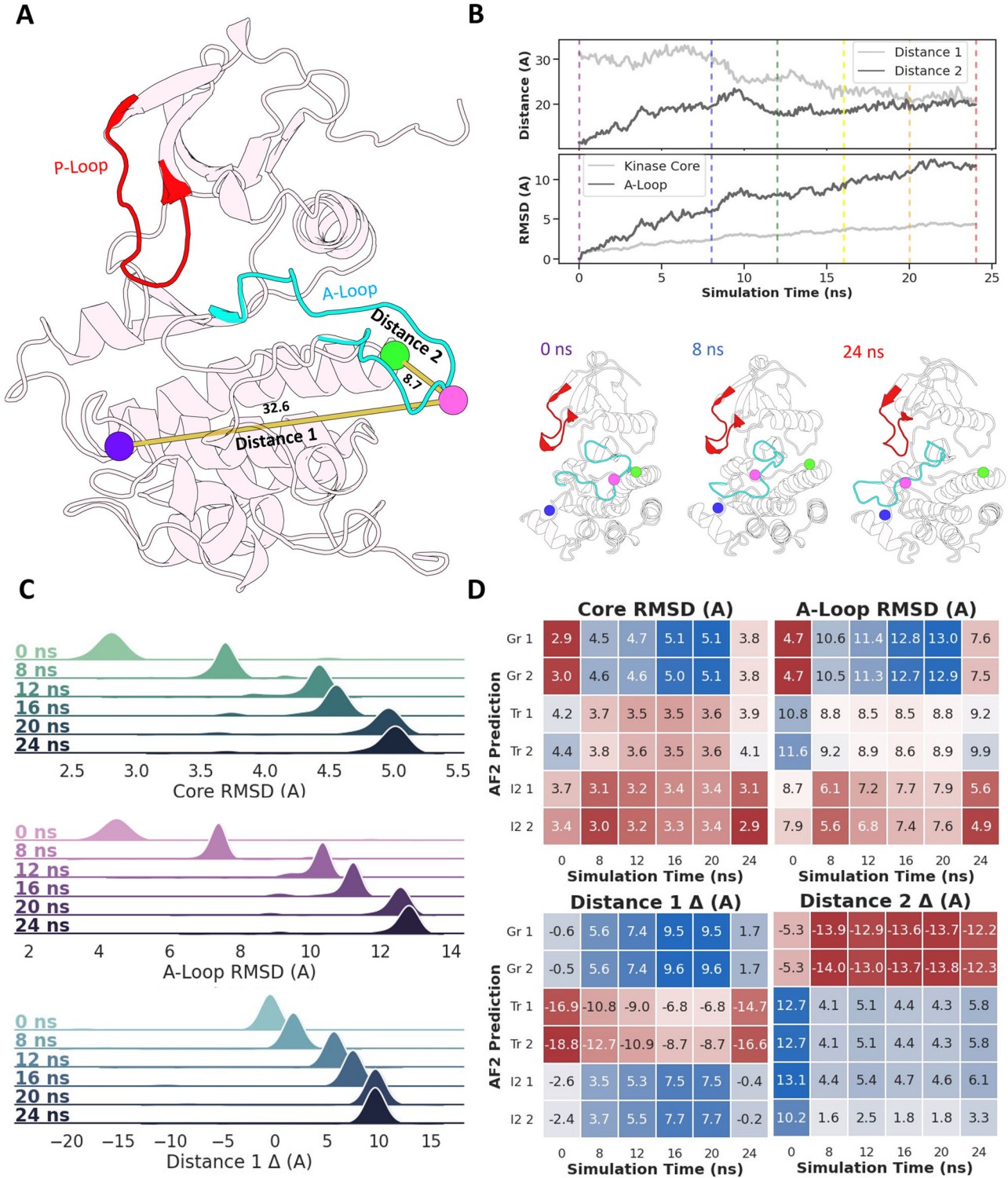

**Fig. 4 | Comparison between the I1 to I2 trajectory obtained using enhanced-sampling MD simulations of the Abl1 kinase core and representative AF2 predictions. A** Structural elements relevant for Abl1 function and that are expected to shift significantly over the I1 to I2 transition. **B** Evolution of four relevant structural observables across the trajectory containing the I1 to I2 transition. Vertical lines indicate representative snapshots extracted from the trajectory for downstream comparisons with AF2 predictions (top). Three representative snapshots from the MD trajectory at 0 (I1), 8 (transition), and 24 (I2) ns, respectively (bottom). **C** Distribution of three observables relative to the MD snapshots for 160 subsampled AF2 predictions. Core and A-Loop RMSDs are defined as the backbone RMSDs of each AF2 prediction's kinase core (residues 242–459) or activation loop (residues 379–395) vs. the kinase core or activation loop backbone of the MD snapshot selected at each time point. Distance deltas are defined as the difference in atom pair distances between each AF2 prediction and each respective MD snapshot. Distance 1 corresponds to the distance between the backbone oxygens of E377 and L409, and Distance 2 corresponds to the distance between the backbone oxygens of L409 and G457. **D** Comparison between six representative AF2 predictions and the six MD snapshots described above. Source data are provided as a Source Data file.

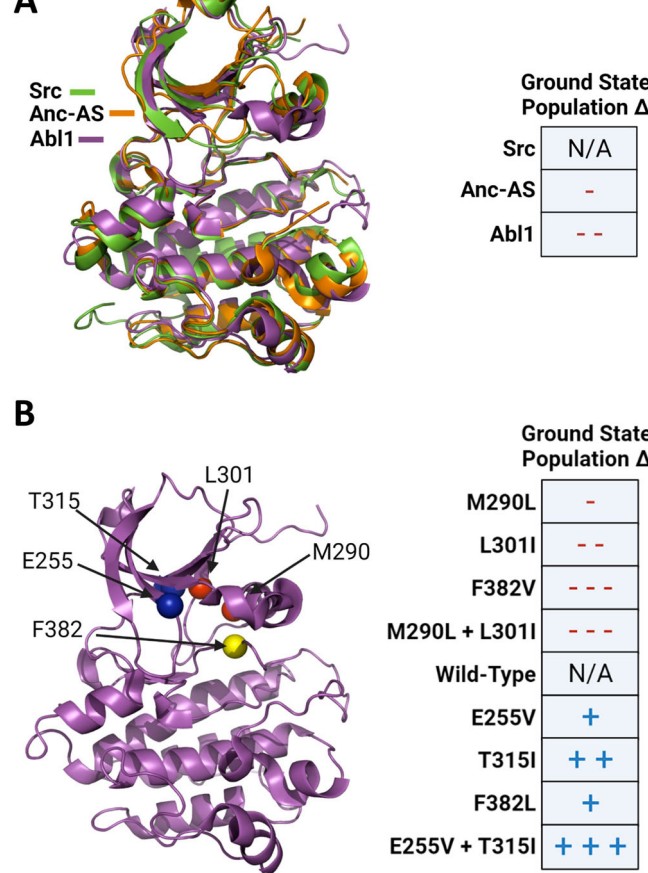

**A**

Src (green)
Anc-AS (orange)
Abl1 (purple)

| | Ground State Population Δ |
|---|---|
| Src | N/A |
| Anc-AS | - |
| Abl1 | - - |

**B**

| | Ground State Population Δ |
|---|---|
| M290L | - |
| L301I | - - |
| F382V | - - - |
| M290L + L301I | - - - |
| Wild-Type | N/A |
| E255V | + |
| T315I | + + |
| F382L | + |
| E255V + T315I | + + + |

**Fig. 5 | Summary of experimental observations regarding the relative state populations of kinase cores along Abl1's evolutionary line and in the Abl1 allelic series studied here. A** Abl1, Anc-AS, and Src are part of the same evolutionary line and share significant homology, but Abl1 occupies the ground state significantly less frequently than Src, and Anc-AS occupies it slightly less than Src, suggesting that Abl1 evolution has directed it to be more flexible than Src[26]. **B** A series of point mutations in wild-type Abl1 are known to increase or decrease the relative population of the enzyme's active (ground) state[23]. Residues with mutations known to increase the population of the ground state are shown as blue spheres in the Abl1 structure on the left side, while those with mutations known to decrease the ground state population are shown as red spheres. The yellow sphere denotes phenylalanine 382, part of the DFG motif, which can be mutated to valine to slightly increase the ground state population, or to leucine to reduce it. While the effects of the E255V + T315I mutation on the ground state population were not directly reported in the literature, we used its cumulative effect on kinase activity and Imatinib binding[55] to infer an increase in the ground state population. Figure Created with BioRender.com.

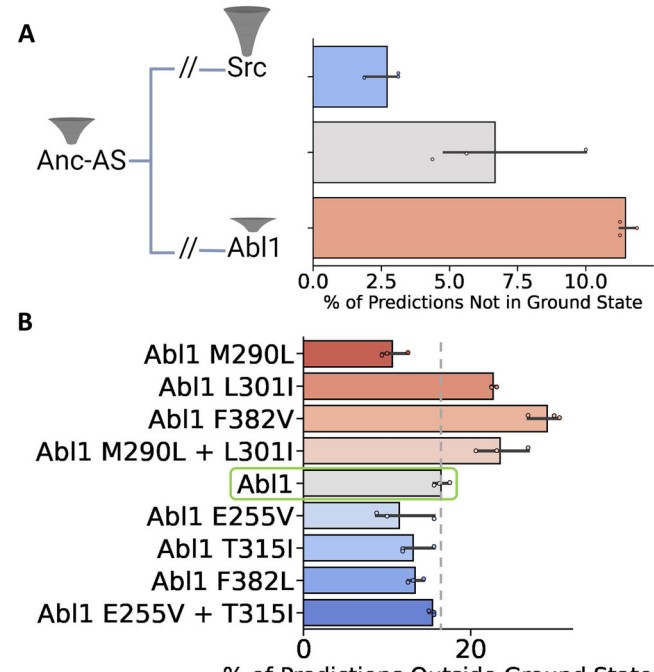

**Fig. 6 | Subsampled AF2 predictions of the percent of conformations not in the ground state for proteins along the Src to Abl1 evolutionary pathway and Abl1 resistance-causing mutations. A** Representation of the evolutionary history connecting Abl1, Anc-As, and Src, and the relative population of the ground state for each kinase core[26]. Right: Percentage of predictions that fall outside of the ground state for each test case. **B** Results of subsampled AF2 for the Abl1 kinase domain allelic series. Data are colored based on the expected relative state populations of each mutation: red-tinted bars represent mutations that are known to decrease the ground state population, while blue-tinted bars represent mutations with the opposite effect[23]. Data are presented as mean values +/− standard error of the mean, and were calculated from three independent sets of predictions, each with 32 unique seeds and 5 models, totaling 160 predictions per replicate. Source data are provided as a Source Data file. Figure Created with BioRender.com.

structures. We then repeated every previous step minus the parameter optimization for each GMCSF mutant and quantified the ground state populations from the resulting set of predictions. Although we were still capable of sampling different conformational states even with GMCSF's shallow MSA, it is worth noting the occasional prediction of partially unfolded conformations with no experimental analogs. The same did not happen with the Abl1 example. We posit that this loss of resolution at aggressive subsampling levels could be a consequence of the shallow input MSA.

To assess how mutations affected the conformational distribution of GMCSF, we measured the RMSD of specific backbone atomic positions of each predicted GMCSF structure with respect to the AF2 prediction that was most similar to the wild-type crystallographic reference (PDB 1CSG)[41]. The regions mentioned above correspond to GMCSF elements known to show significant perturbations in NMR experiments. The results of this analysis for the two regions with the most significant changes in our NMR results are described in Fig. 8. Additionally, we also measured the distance between the alpha carbons of H15 and H83 as well as the overall backbone RMSD with respect to the ground state reference for each prediction with the goal of identifying predictions that led to unexpected conformations or partially or completely unfolded structures. These additional measurements as well as three examples of unexpected structures are illustrated in Supplementary Fig. 14.

Despite the paucity of sequence data, our subsampled AF2 method correctly identified residues H15 and H83 as the most sensitive

Accordingly, we sought to test if our subsampled AF2 method was capable of predicting the expected changes in the backbone dynamics of each mutant with respect to the flexibility of helix A and other structural elements in GMCSF. Our predictions were considered accurate if they matched NMR experiments we performed along two axes: if they predicted mutations to H15 and H83 to provoke larger changes in the distribution of conformations than mutations to H87; and if they correctly predicted the most significant mutations in H83 or H87 as evidenced by CSPs or broadened peaks. After building the MSA using the wild-type human GMCSF sequence as a query and the JackHMMR method[32], we determined the max_seq and *extra_seq* parameters that led to the greatest diversity of GMCSF conformations (see Supplementary Fig. 13).

Employing these parameters while keeping all others the same from the Abl1 tests, we used AF2 to predict wild-type GMCSF

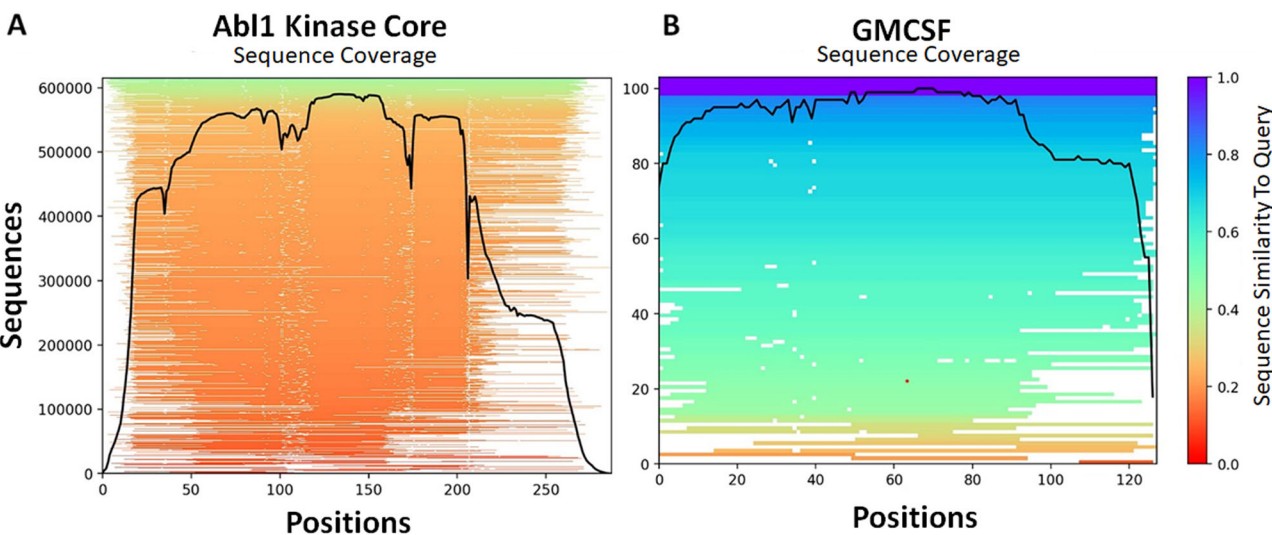

**Fig. 7 | Comparison between MSA length and per-position coverage for two protein systems whose conformational ensembles were predicted in this study.** **A** Abl1 kinase core. **B** GMCSF.

to mutation, as the range of the distribution of RMSDs of residues 80–90 and 110–125 is significantly larger for most of the mutations tested at both of these sites. Conversely and in line with the experimental results, mutations to H87 led to significant changes in the distribution of the residue 80–90 RMSDs, but to comparatively modest changes for the RMSDs of residues 110–125. In addition to accurately predicting the differences in amplitude of backbone rearrangements between mutants H15/83 and H87, subsampled AF2 correctly estimated the significant impact of mutations H83R and H83N on C-terminal conformations, while also accurately predicting H83N, H83Y, and all three H87 mutants to have a large impact on the R80–90 RMSD distribution.

As with the Abl1 allelic series example, our method did not achieve perfect accuracy. Specifically, it failed to predict the significant changes in GMCSF's N-terminus dynamics associated with the H83N mutation (see Supplementary Fig. 11). Additionally, our method failed to replicate the significant changes in dynamics of residues 80–90 for the H83Y mutations, which should be larger in amplitude than those for H83N and closer in amplitude to those for H83R. Similarly, we expect the H87Y mutation to induce larger conformational changes in the region composed of residues 80–90 than the H87N mutation due to the enrichment of broadened peaks in that region in the H87Y NMR results, but AF2 predicts H87Y to be significantly less influential than H87N in that context. In summary, AF2 performed exceptionally well at predicting subtle conformational changes in specific loops for most of our test cases but failed at replicating the comparatively larger expected effects of mutation H83Y. Refer to Supplementary Table 5 for a measurement of the statistical power of comparisons between ensembles of predictions belonging to wild-type or mutant GMCSF.

Beyond these limitations, we also observed a set of alternative conformations that are significantly different than both the ground and open states. Clustering the results by the structural features described in Supplementary Fig. 15 reveals that one alternative conformation, in particular, is significantly enriched, especially in predictions of the H83 mutants. In this alternative conformation, henceforth dubbed A1, the C helix has switched places with the B helix, placing H83 and H15 more than 10 Å away (Supplementary Fig. 14A). In this state, helix B occupies the groove to which heparin is thought to bind, which could be a mechanism for self-inhibition. We believe that further NMR experiments such as chemical exchange saturation transfer (CEST) or ensemble studies via residual dipolar coupling could be the best way to confirm if this is a metastable GMCSF conformation with

physiological functions. Although certainly promising, these experiments are beyond the scope of this study due to their complexity and cost. In the absence of ground-truth evidence, the facts that these predictions display structural rearrangements with amplitudes that seem to be comparable to those of the H83 mutations (as evidenced by the H83 NMR chemical shift perturbations and broadened peaks [Supplementary Fig. 11]) and that these conformations were predicted more often for H83 mutants highlights the promise of using subsampled AF2 to help understand NMR results and derive novel hypotheses or mechanisms.

Finally, many of the alternative conformations not discussed above were found to be partially unfolded. Although the frequency of these was low compared to predictions binned to the ground or A1 states, the fact that they existed at all was surprising as running subsampled AF2 for the Abl1 test case led to no unfolded predictions whatsoever. One plausible hypothesis beyond the destabilizing effects of the mutations that may explain this observation is that the substantially shorter length of the GMCSF MSA relative to Abl1's drastically increases the uncertainty in AF2's predictions leading to a wider range of predictions.

### Additional examples

To shed light on how well our results on the Abl1 kinase core and GMCSF generalize and how the accuracy of our predictions depend upon MSA depth and diversity, we recognize the need for more tests on a wider array of protein systems. We have therefore curated an additional test set composed of eight proteins diverse in size, dynamics, function, and evolutionary history. The results of this challenge are thoroughly described in the Supplementary Appendix (Supplementary Table 4, Supplementary Figs. 16–27), and ultimately support the notion that the approach described in this study can be applied for predicting the conformational landscapes of a wide array of proteins and their variants.

## Discussion

In this work, we successfully modified AF2's inputs and parameters to predict the conformational ensembles and relative state populations of two proteins, Abl1 kinase and human granulocyte-macrophase colony-stimulating factor (GMCSF), that have very different amounts of known sequence homology. In studying these proteins, we focused on how well our subsampled version of AF2 can reproduce the effects of evolution and mutations on dynamics. In addition to these two clearly

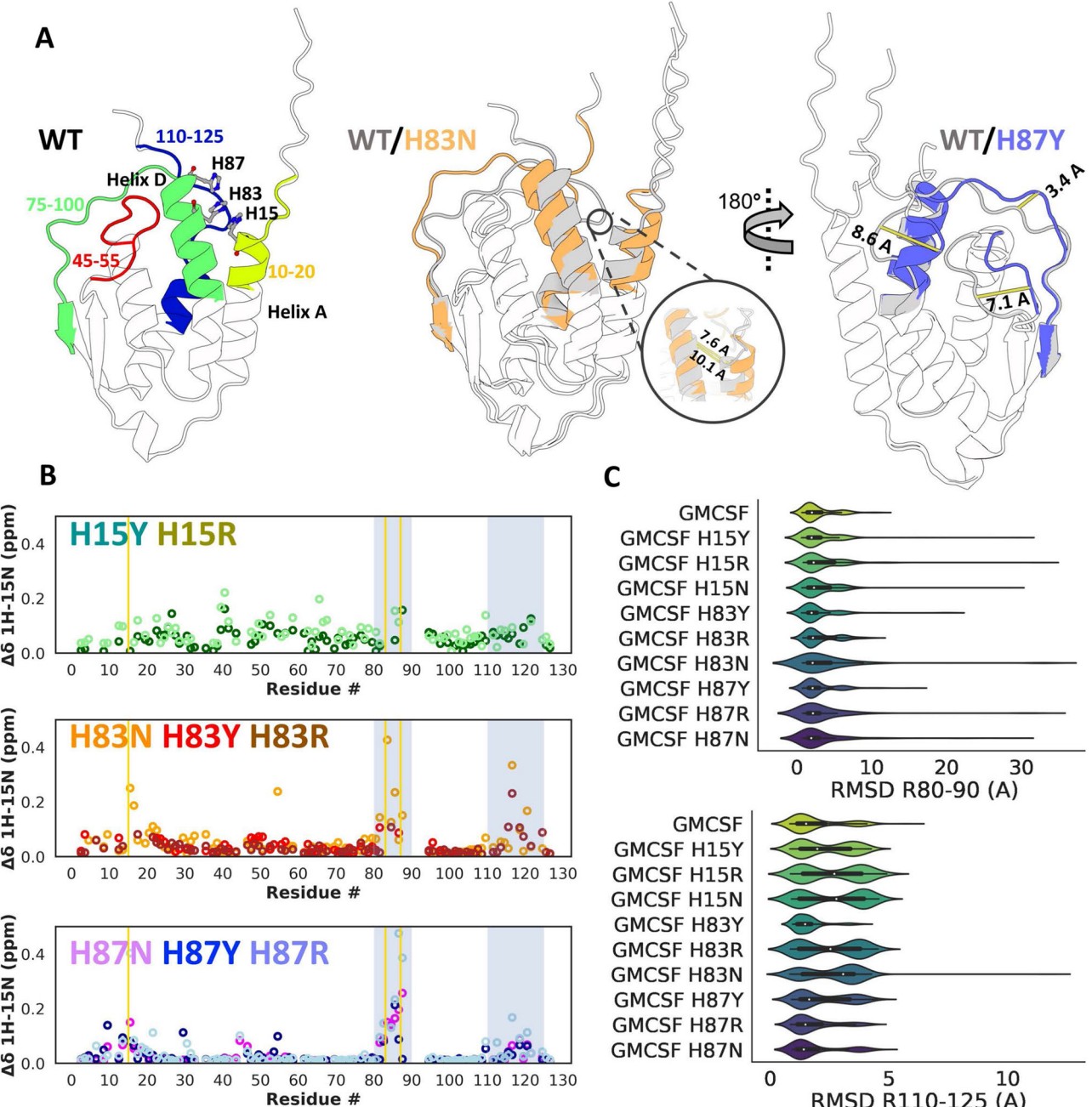

**Fig. 8 | Subsampled AF2 results for GMCSF mutations. A** Left: Annotated wild-type GMCSF structure in the ground (closed) conformation as predicted by AF2. Each colored element represents a region of putative high mobility according to chemical shift perturbations (CSPs) in the histidine triad mutants, identified by visual analysis of the CSP data. Specifically, residue ranges with significant perturbations and peak broadening were designated as regions of putative high mobility. Middle: Superposition of GMCSF in the ground conformation and a putative open conformation whose population is enriched in the AF2 predictions for H83 and H87 mutants, especially H83N. The inset compares distances between codon positions 15 and 83 in both conformations. Right: Superposition of GMCSF in the ground conformation and a tilted helix D conformation whose population is enriched in the AF2 predictions for H87 mutants (especially H87Y). Distances between flexible elements for both conformations are shown as yellow bars. **B** Per-residue chemical shift perturbations for GMCSF mutants, separated by codon position. Gold vertical bars represent the three mutation sites, and silver shaded areas correspond to the residue indices plotted on the x-axis in **C**. **C** Distributions of root mean square deviations of atomic positions for the backbone atoms of residues 80–90 (top) and 110–125 (bottom) superimposed upon the GMCSF wild-type ground state structure. The distributions span 480 independent predictions (32 unique seeds * 5 models * 3 replicates. The center of the inset box plot is defined as the median of each dataset, and corresponds to the following values for the R80–90 range: 2.11, 2.26, 2.41, 2.44, 2.04, 2.39, 2.01, 2.23, 2.20, 2.01; and to the following values for the R110-125 range: 1.84, 2.16, 2.52, 2.56, 1.72, 2.39, 2.65, 1.79, 1.73, 1.70. Source data are provided as a Source Data file.

contrasting examples, we also measured our approach's capacity for predicting alternative conformations and their relative populations (when known) for a diverse set of eight protein systems.

Our subsampled AF2 implementation predicted Abl1, a kinase for which there is an abundance of sequence homology, to occupy its ground state nine times more frequently than its I2 state, consistent with NMR observations. More importantly, subsampled AF2 correctly predicted the relative state populations of two evolutionarily related kinases, Anc-AS and Src kinase, leading to the correct correlation between their evolutionary distances and relative state populations.

Using single and double Abl1 mutants with known effects on the relative state populations of Abl1's ground and I2 states, we moreover found that AF2 yielded surprisingly accurate state populations even for single mutations. This is best evidenced in the results for the F382 substitutions: phenylalanine 382 is a codon known to slightly increase the population of Abl1's ground state if mutated to leucine, or significantly reduce it if mutated to valine, an observation that is accurately replicated by our prediction method. Furthermore, an unexpected but remarkable feature illustrated by all of the Abl1 test cases is the capacity of subsampled AF2 to predict intermediate conformations spanning the transition from the ground state to I2 which closely match intermediate conformations obtained from more costly MD simulations.

We also obtained overwhelmingly accurate conformational ensembles that correlated with NMR experiments for GMCSF. Despite the lack of GMCSF sequence data which led to an input MSA of fewer than 120 sequences (versus more than 600,000 for Abl1), predictions from subsampled AF2 were distributed with frequencies that matched observed changes in dynamics for most GMCSF variants. These results suggest that AF's prediction engine is robust enough to decode some conformational distribution information from relatively scarce data.

The results we have obtained for the two test cases discussed above confirm AF2's potential for predicting conformational ensembles and, more importantly, demonstrate unforeseen applications of AF2. In particular, we show that optimization of subsampling parameters allows users to accurately predict relative state populations and how they change in response to mutations with single-substitution resolution. This feature is a significant advance over the previous state of the art, as it facilitates high-throughput applications such as the design of fold-switching proteins, inference of mechanisms of acquired drug resistance, and reweighting of binding affinity predictions. Additionally, our workflow generates conformational intermediates, which has direct implications for discovering drugs that bind alternative conformations and for improving our understanding of biophysics in general. In addition to these immediate practical applications, the more fundamental observation that AF2 is decoding information regarding conformational distributions from sequence data alone points to many other potential unforeseen uses of AF2 that can result in further methodological advances and discoveries.

This said, our AF2 pipeline is not perfect; our workflow inaccurately predicted the M290L mutation to significantly decrease the ground state population when that mutation is known experimentally to have the opposite effect. Interestingly, AF2 predicted the double mutation (M290L + L301I) to increase the ground state population more than the L301I mutation alone. As AF2 correctly predicted the relative state populations of Src which differs from Abl1 by dozens of mutations, this suggests a potentially more general trend that AF2 is more accurate in predicting the effects of multiple mutations than those of a single mutation.

Furthermore, our pipeline also struggled to correctly predict all of the structural elements expected to differ in each conformation. Specifically, while AF2 predicted an ensemble of activation loop conformations, a few of the inactive-like predictions (activation loop closed) contained structural elements that are typically thought to belong to the active state, such as the position of the phosphate-binding loop and the dihedral angles of D381 and F382. Moreover, even when our pipeline predicts a change in a structural element from its configuration in the ground (active) state, the amplitude of the change may not correspond to that seen in experimentally-resolved structures. For example, the side-chain dihedral angle of the D381 residue in Abl1 with the plane formed by the A380 side-chain ranges from −130 (active state) to 40 (inactive state) degrees in structures in the Protein Data Bank (PDB)[23], but ranges from −130 to −90 degrees in AF2 predictions. Last but not least, while our method is capable of predicting whether mutations will increase or decrease state populations, it

remains to be shown whether it is capable of directly predicting Boltzmann ratios of states quantitatively.

The diverse protein test set also allowed us to evaluate how our modified AlphaFold2 approach fared at predicting conformational changes of different scales, both in terms of the number of atoms involved and in the expected timescale of the change itself. From this analysis, we observed that subsampled AlphaFold2 fared better at predicting large and slow conformational changes, such as the LmrP or LAT1 channel openings that involve the correlated motions of hundreds of backbone atoms (Supplementary Figs. 17 and 20, respectively). For these types of conformational changes, AlphaFold2 predicted a variety of potentially intermediate conformations spanning the transition and both ground and high-energy states for a wide range of subsampling values. For faster or less significant conformational changes, such as the conversion between the Fyn-SH3 intermediate and its folded ground state (Supplementary Fig. 23), subsampled AF2 predicted no potential transition conformations and only predicted both states under narrow subsampling conditions, suggesting a resolution limitation.

Additionally, subsampled AF2 performed better at predicting alternative conformations when used on systems whose high-energy states are more frequently populated. For example, the NMR-resolved relative populations of wild-type Abl1's and mutant Fyn-SH3's higher-energy states are 6% and 2%, respectively. While our approach successfully predicted wild-type Abl1's (I2, inactive) high-energy state using multiple subsampling conditions, our approach was only able to predict the high-energy intermediate folding state of mutant Fyn-SH3 within a narrow range of subsampling conditions. Further, the Abl1 ensemble of predictions often included a small but significant (>5%) population of putative in-between conformations, while the mutant Fyn-SH3 prediction ensemble did not include such structures (Supplementary Fig. 23). These results indicate that there is some minimum threshold for the relative population between 2 and 6% to be able to detect higher energy states using subsampled AF2.

Although these results are overwhelmingly positive, they also come with caveats that were not observed in the Abl1 example. Notably, our capacity to predict relative state populations without prior knowledge was limited for a few protein systems within the test set, such as LAT1, for which the ground state population was dependent on the level of subsampling. Additionally, in some examples such as AkrB, aggressive subsampling led to the prediction of unfolded conformations unlikely to map to the functional states of the protein studied. In the Supplementary Appendix, we discuss a few heuristics for minimizing the incidence of these potential pitfalls, such as measuring the coverage of the putative pathway between two presumed states and estimating the frequency of unfolded predictions. Although it is not within the scope of this work to offer a one-size-fits-all approach for discovering appropriate subsampling parameters for every protein system, we believe that these suggestions can help orient future studies seeking to accomplish similar objectives.

Despite these limitations, we believe that our results showcase promising unexpected applications of AF2. Using the high-throughput pipeline we developed or one inspired by it could save significant time when filtering large allelic series to identify mutations with significant impacts on conformation for further study using NMR or other, more expensive experimental or computational techniques. It could also be used as a prediction engine for classifying arrays of drug-resistant mutations based on their shared effects on the stability of a given conformation, thus facilitating polypharmacology. Finally, as demonstrated by AF2's Abl1 activation loop predictions, our method could also be useful for identifying previously unknown, potentially meta-stable states of known proteins. While it remains to understand exactly how AF2 is gathering and interpreting signals about state populations from sequence data, we hope that our work will motivate many future investigations.

**Table 1 | Summary of parameters used for enhanced-sampling molecular dynamics simulations of Abl1 inactivation**

| Box dimensions | (112.4, 124.3, 118.0) |
|---|---|
| Total number of atoms | 77251 |
| Number of water molecules | 724982 |
| Salt concentration | 0.125M |
| Thermostat | Noose-Hoover[54] |
| Barostat | Monte Carlo barostat[44] |
| Nonbonded cutoff | 10.0 A |
| Temperature | 300 K |
| Pressure | 1 ATM |
| Prod. Integration Timestep | 2 fs |
| t (W.E. iteration length) | 100 ps |

## Methods

### Protein structure visualization
To visualize the predicted structures and trajectories and calculate descriptors such as distances between atoms, RMSD to reference, dihedral angles, etc., we used PyMol (version 2.4.1) (Schrodinger LLC, 2020).

### Multiple sequence alignments
The JackHMMR algorithm[32] was used to generate multiple sequence alignments (MSAs) for each protein of interest by querying sequences from the UniRef90[33], Mgnify[35], and small BFD[34] databases.

### Structure prediction
We used AlphaFold 2[11] within the localcolabfold colabfold-batch 1.5.0 implementation[42] to predict protein structures of Abl1, Src, ANC-AS, and of GMCSF.

### Molecular dynamics simulations
Enhanced-sampling molecular dynamics simulations were conducted on the proteins described using the WESTPA2[43] implementation within the OpenMM molecular simulation engine[44]. The general parameters for the simulations are described in Table 1. For an extensive description of the MD methodology employed in this study, please refer to the Supplementary Materials.

### Protein expression and purification
Plasmid DNA containing either wildtype GMCSF or GMCSF containing a point mutation with an N terminal 6-His tag was cloned into a pET-15b vector and transformed into BL21(DE3) cells in a manner described elsewhere. Isotopically enriched GMCSF was expressed at 37 °C in M9 minimal medium containing $CaCl_2$, $MgSO_4$ and MEM vitamins with $^{15}NH_4Cl$ as the sole nitrogen source. Small cultures of GMCSF were grown overnight in LB medium. The following morning, cloudy suspensions were collected by centrifugation and resuspended in the final M9 growth medium. Cultures of GMCSF were grown to an $OD_{600}$ of 0.8–1.0 before induction with 1 mM isopropyl $\beta$-D-1- thiogalactopyranoside (IPTG). Post induction, cells were kept at 18 °C while shaking. Cells were harvested after 18h and resuspended in a denaturing lysis buffer containing 10 mM Tris-HCl, 100 mM sodium phosphate, and 6 M guanidine hydrochloride (GuHCl) at pH 8.0. Cells were lysed by sonication and cell debris was removed by centrifugation. The resulting supernatant was incubated and nutated with 10 mL of Ni-NTA agarose beads for 30 min at room temperature before the Ni-NTA slurry was packed into a gravity column. The column was washed with the initial lysis buffer, followed by a gradient of the same buffer without GuHCl over 100 mL. Elution of GMCSF in its denatured form was performed with 1 column volume of a buffer containing 10 mM Tris- HCl, 100 mM sodium phosphate, and 250 mM imidazole at pH

8.0. GMCSF was refolded by dilution via dropwise addition of the 10 mL elution into 100 mL of a refolding buffer containing 10mM Tris-HCl, 100 mM sodium phosphate, and 750 mM arginine at pH 8.0. The refolded protein was dialyzed exhaustively against a buffer containing 2 mM sodium phosphate at pH 7.4. GMCSF was concentrated to ~200 µM with an Amicon centrifugal device and stored at −20 °C.

### NMR spectroscopy
NMR samples were prepared by dialyzing 200 µ GMCSF and GMCSF mutants against a buffer of 20 mM HEPES and 1 mM EDTA at pH 7.4. NMR experiments were performed on a Bruker Avance NEO 600 MHz spectrometer at 25 °C. NMR data were processed in NMRPipe[45] and analyzed in Sparky[46]. The $^1H$ and $^{15}N$ carrier frequencies were set to water resonance and 120 ppm, respectively.

All NMR experiments were performed using a Bruker Avance NEO 600 MHz spectrometer equipped with a triple resonance inverse detection cryoprobe. The data were collected through the companion software Topspin 4.0.3 using a 2D HN correlation via double inept transfer heteronuclear single quantum coherence (HSQC) pulse program (hsqcetfpf3gpsi) at 298 K[47–50]. Spectral processing was performed using NMRFAM-SPARKY[46] while data analysis was performed using Microsoft Excel 2021[51] and GraphPad Prism 10.0.1 for MacOS[52]. All GMCSF NMR samples were stored in 20mM HEPEs 1 mM EDTA, at concentrations between 300–500 uM.

### Miscellaneous and data visualization
Data plotting and visualization were performed using Seaborn (version 0.11.1).

Figures were composed with BioRender (BioRender.com, 2020).

### Reporting summary
Further information on research design is available in the Nature Portfolio Reporting Summary linked to this article.

## Data availability
Sample MSAs and the resulting analysis of AlphaFold2 predictions and MD simulation data generated in this study have been deposited in the GitHub repository https://github.com/GMdSilva/gms_natcomms_1705932980_data[53]. Due to storage limitations, the repository neither includes the resulting PDB ensembles nor all of the MSAs generated and used in this study, although access can be obtained by contacting the corresponding author. Source data are provided with this study.

## Code availability
Scripts used to assemble the MSAs and run AlphaFold2 with different subsampling conditions have been deposited in the GitHub repository https://github.com/GMdSilva/gms_natcomms_1705932980_data[53].

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

## Acknowledgements

The authors thank Jacob Rosenstein, Joseph Larkin, F. Marty Ytreberg, Jagdish S. Patel, Marcelo D. Polêto, Winston Y. Li, Haibo Li, and Kyle Lam for insightful conversations and support. G.M.d.S. and J.Y.C. were supported by a Blavatnik Family Fellowship award. G.M.d.S. and B.M.R. were supported in part by the National Science Foundation under Grant No. 2027108. J.Y.C. and G.P.L.'s contributions were supported by NIH Grant R01 GM144451. The computational aspects of this research were conducted using computational resources and services at the Center for Computation and Visualization, Brown University.

## Author contributions

G.M.d.S. performed the simulations and analyzed the sequence and AlphaFold 2 data. J.C. performed NMR experiments and contributed NMR data. D.C.D., G.P.L., and B.M.R. provided direction and oversight. G.M.d.S. and B.M.R. drafted the manuscript. All authors provided notes and edits to the manuscript.

## Competing interests

The authors declare no competing interests.
