## [Peer Review File · Nature Communications]

High-throughput Prediction of Protein Conformational Distributions with Subsampled AlphaFold2Reviewer #1 (Remarks to the Author):

This is a potentially interesting paper that aims at presenting a methodology based on AlphaFold2 to develop conformational ensembles. However, as written the manuscript is unclear, and further it is difficult to tease out how the input parameters need to be adjusted in order to achieve the optimal amount of subsampling that results in a large enough ensemble that is not too disordered. What the authors need to present is additional examples on other well characterized systems that exist in multiple conformations (I can name calmodulin, cyclophilin A, Alkb, SH3, Enzyme I, ... but there are several other examples in the literature) and discuss how the agreement between predictions and experimental data depends on the amount of subsampling. In addition, other important points to discuss are: 1) how populated the high-energy states need to be in order to be detected, and 2) how well the method performs on larger conformational transitions. The second point is of primary relevance, since AlphaFold is notoriously challenged by multidomain systems that undergo large-scale conformational changes (see for example DOIs 10.1021/jacs.3c03425, 10.1002/pro.4175, and 10.1016/j.jmb.2021.167336).

Other minor points to address:

- 1) Please provide a detailed description of how the ensemble members were assigned to the ground state or to different excited states.
- 2) How were the "regions of putative high mobility" determined based of the chemical shift perturbation data in Figure 8?
- 3) Please plot the chemical shift perturbation data from Figure S7 as a color gradient on the atomic resolution structure. Also, acknowledge that these chemical shift changes (as well as the observed line-broadening) may originate from conformational changes involving the surrounding side chains, and that other experiments (i.e. RDC) should be run to confirm that the chemical shift perturbations are the result of changes in backbone dynamics.
- 4) Did the author make any attempt to measure the relative populations of ground and excited state in GMCSF by NMR? These values would make the comparison with the computational predictions much more stringent.
- 5) Please discuss best practices to test predictions experimentally.

Reviewer #2 (Remarks to the Author):

In this paper, the authors explore the use of both MSA subsampling and dropouts to investigate if AlphaFold2 can be used to estimate population sizes. The results look promising and take this approach a small step further than earlier studies. However, the study is extremely small (only three examples) so it is not clear to know if this better provide any advantage over earlier subsampling (e.g. by Meiler) or dropout methods (Wallner).

Major

The study should be expanded significantly. At the bare minimum all models tested by del Alamo et al should be included. But it could probably be expanded as well. Note that this will not be a reproduction of earlier results, as another AF version was used in their studies.

A negative set (i.e. proteins with a single conformation) is absolutely necessary to include. It is not obvious how to test this, but I would argue that superfamilies with many known protein structures and very little structural variation could form such a negative set.

Statistics is needed, i.e. a comparison of variation for proteins assumed to have multiple conformations vs. others. How is this biased using different subsampling techniques (values) (as in Fig 1A but on a larger scale), with and without dropouts etc.. It would also be good to include other versions of AlphaFold as they behave differently.

Reviewer #3 (Remarks to the Author):

Predicting Relative Populations of Protein Conformations without a Physics Engine Using AlphaFold 2

The authors showed how to use AlphaFold 2 to directly predict the relative populations of the two proteins under consideration. They tested the approach on (1) Abl1 kinase and (2) granulocyte-macrophage colony-stimulating factor and predicted their conformations.

[1] The abstract is a bit too positive given that it makes no mention of the issues that the method encountered, such as the inability to create samples within the Abl1 inactive state 2 (I2), mis-ranking of some mutants, non-quantitative interpretability of the ranking of the mutant, and similar issues with granulocyte-macrophage colony-stimulating factor (GMCSF). Understandably, all of these cannot be put in the abstract, but some of the issues should. It is left to the authors to decide which ones to include.

[2] The authors mentioned that they systematically tested the "accuracy of different AF2 parameter combinations for predicting the Abl1 kinase core structural ensemble." Were the same set of parameters used for both protein systems studied? How can one be sure that the method presented in the paper is not overfitted to Abl1 kinase or to the granulocyte-macrophage colony-stimulating factor? How should one prove that the method is generalizable to all proteins?

[3] The authors mentioned that they compiled an extensive MSA spanning over 600,000 sequences. How did the authors determine that 600,000 is enough? Will the approach work with a much smaller collection of sequences? Given that the authors also showed the results for GMCSF where they used only about 120 sequences, the authors should comment on the number of sequences for GMCSF affected its results if it did. If it did not, why not? Did the authors experiment with what the results would be if the number of sequences was smoothly varied, say from 600,000 to 30?

[4] The authors mentioned using dropouts (and 32 predictions with independent seeds for each) during inference to sample from the uncertainty of the models. The authors should elaborate. Were the dropout configurations identical to the ones initially used to train the model? Readers would want to know. In addition, what would be the effect of higher dropout probability or lower dropout probability on the distribution of the conformation space of the protein sampled?

[5] The authors mentioned that "max_seq:extra_seq ratio of 256:512 leads to the most diverse results in terms of activation loop conformations." Do you really mean ratio or these specific numbers? For example, since 256:512, 100:200, 512:1024, ..., are equivalent, would all of them have worked the same way and produce comparable/identical results?

[6] The authors mentioned that "none of the Src predictions were found to be in the I2 state, although the enzyme is known to infrequently occupy this conformation. This suggests a resolution limitation in using AF2 to predict relative state populations: conformations with very low occupancy such as I2 in Src might be missed by the algorithm in its current implementation." Would fine-tuning AF2 with new sequences / additional data help? It may help the readers to comment further on these.

[7] Are these all the mutations with known effects on the population of the ground state? If not, why did the authors test these mutations alone? Does the method work for the other mutations (whose effects are known) too?

[8] In Figure 8A, the authors should add a vertical line to the 180o rotation arrow. The line should help the readers to know the axis of rotation.

REVIEWER COMMENTS

First of all, we thank the reviewers for taking the time to thoroughly review and comment on our manuscript. We have taken many of the comments about providing more examples to facilitate generalization to heart and have since provided many new examples in the Supplementary Materials. All of the suggestions provided have improved the quality of our manuscript. In what follows, we do our best to address your feedback below.

Reviewer #1 (Remarks to the Author):

1) What the authors need to present is additional examples on other well characterized systems that exist in multiple conformations (I can name calmodulin, cyclophilin A, Alkb, SH3, Enzyme I, ... but there are several other examples in the literature) and discuss how the agreement between predictions and experimental data depends on the amount of subsampling.

A: We agree with the assessment that more examples illustrating how our AlphaFold2 subsampling approach can capture the conformational distributions of different, well-studied proteins would significantly increase the impact of our study. We thank the reviewer for suggesting specific systems for exploration.

To address this important suggestion, we have applied our prediction heuristic to eight new systems in total, with the results outlined in Appendix 2 of the Supporting Information, Figures S16 to S27, and related data placed in our GitHub.

The new test set is distributed as follows:

1. Proteins suggested by Reviewer 1: Calmodulin, AlkB, and Fyn SH3
2. Proteins suggested by Reviewer 2: LAT1, LmrP, and CCR5 (three samples from the del Alamo study), and Carbonic Anhydrase VI (negative control, a knotted enzyme with very slight intrinsic dynamics)
3. Protein identified by us as an interesting test case: Aurora Kinase A

These eight systems display a large variety of functions, protein sizes, and evolutionary histories. For all of the tested proteins, we applied our prediction heuristic with nine subsampling parameter combinations ranging from 4:8 to 1024:2048, with $n=480$ (five models times 96 seeds) for each, totalling over 30,000 individual AlphaFold2 predictions for the entire set.

To assess the accuracy of our method in predicting alternative states and in-between conformations of proteins contained in the new test set, we measured the similarity of each prediction in the ensemble to references of known states in the literature, and plotted the distribution. When that information was known, we also evaluated our method's capacity for accurately predicting the ground state to be the most populated conformation.

In all cases besides the negative control (Carbonic Anhydrase VI), we identified subsampling parameters that correctly led to the prediction of at least two well-characterized conformational states, and in most cases, a collection of transitional conformations between them. Crucially, our approach succeeded in predicting major states in the correct populations

and transitions for test systems CCR5 and LmrP, which was not possible with previous methods such as from del Alamo and collaborators [1]. Building upon our Abl1 and GMCSF results, we also successfully predicted the effects of point mutations in the conformational landscapes of two more systems (Calmodulin and Fyn-SH3).

We conjecture that our approach's improved predictive power compared to previous methods stems from the decision to build deep MSAs with jackhmmer, which we believe to significantly boost the coevolutional signal pertaining to dynamics. To demonstrate this difference, we compared wild-type Abl1 kinase domain ensembles predicted with different MSA depths and origins (mmseqs2 or jackhmmer, or truncated jackhmmer). This analysis, highlighted in Figure S18, revealed that it is not merely MSA depth that leads to our increased power to predict major conformational states and their intermediate forms, but also the MSA entropy, as the top ~30,000 sequences from the jackhmmer MSA (n~600,000) led to significantly more "in-between" predictions than predictions made from the mmseqs2 MSA (n~30,000). Notoriously, sequences in the truncated jackhmmer MSA are significantly more similar to each other than the sequences in the mmseqs2 MSA. We also repeated this test for the CCR5 and LmrP examples, in which using a deeper/more conserved MSA from jackhmmer proved crucial for better sampling of alternative conformations (Figure S19).

Contrary to the Abl1 or GMCSF cases, for many of the examples in the new test set, the stability of the ground state populations varied significantly based on the level of subsampling employed. This suggests that further analysis might be necessary in some cases in order to identify the best subsampling parameter sets. Although it is not within the scope of this study to develop a one-size-fits-all heuristic for choosing parameter sets, we observed that parameter sets that led to accurate ground state population predictions, in many cases, had strong coverage of in-between conformations and few outlier conformations. Thus, we suggest starting by analyzing coverage as a metric for further scoring candidate subsampling conditions.

Additionally, for the Calmodulin and Fyn-SH3 systems, we also predicted the effects of previously-studied point mutations on the relative populations of the predicted conformations, with qualitative accuracy predicting the direction of each mutation's effect. These results are showcased in Figures S21 and S22 for Calmodulin and Figure S23 for Fyn-SH3.

Finally, we used the Kruskal-Wallis H-test to measure the significance of the differences we observe in each prediction involving mutant vs. wild-type results. This test was chosen because it is unlikely that conformational observable values will follow a normal distribution considering the nature of energy landscapes (which are more likely to exhibit multimodal distributions).

We believe that the inclusion of this diverse and extensive example set addresses the main points raised by Reviewers #1 and #2 with regards to the general applicability and the statistical power of our results.

2) how populated the high-energy states need to be in order to be detected

A: The population of the I2 state in Abl1 is expected to be around 6% [2], and it is predicted in similar frequencies by our subsampled AlphaFold2 approach. Conversely, Src is predicted to occupy the I2 state less frequently than Abl1 (Src I2 population < 6%), but our heuristic leads to no predictions of Src in the I2 state within the resulting ensembles (n=480) using the same subsampling conditions as those used for Abl1.

Interestingly, in the Fyn-SH3 triple mutant test case, we were also able to predict a conformational state that is similar to the intermediate folding state identified in the literature, known to have a relative population of ~2% in solution [3]. However, aggressive subsampling was required to predict this state, and it was not present in the Fyn-SH3 wild-type ensemble.

This suggests that limitations in coevolutionary signal, potential training biases in AF2, and perhaps other factors also play a role in addition to the relative population of a high-energy state. Also importantly, the high-energy states must be predicted with significant frequency in the ensemble so that their population can be measured with statistical robustness, especially for comparisons with mutants. Combined, these caveats prevent us from ascribing a definitive resolution for the minimum population of a high-energy state that can be predicted through our subsampled AF2 approach. In the context of our study, however, that value was ~6% for Abl1, and ~2% for Fyn-SH3 triple mutant.

3) How well the method performs on larger conformational transitions. The second point is of primary relevance, since AlphaFold is notoriously challenged by multidomain systems that undergo large-scale conformational changes (see for example DOIs 10.1021/jacs.3c03425, 10.1002/pro.4175, and 10.1016/j.jmb.2021.167336).

A: For the sake of limiting confounding variables and keeping our model tractable, we have not included multidomain systems in our analyses. That being said, we observe significant conformational transitions in the LAT1 and LmrP cases described above, for which our subsampled AF2 approach is capable of correctly predicting both the outward-open and the outward-closed conformations, as well as a few occluded conformations, essentially capturing the opening/closing pathway, which involves a significant conformational change.

These results are showcased in Figures S17 and S20 of the newly-added Appendix S2. Additionally, we also observe strong correlations between the positions of separate structural elements known to be shifted in tandem in different states. This is noted in the Abl1 test case in which the position of the aC-Helix and the Activation Loop are frequently correlated (as they should be [2]) in predictions classified as either the Ground or I2 states, as observed in the comparison with results from enhanced sampling molecular dynamics simulations (Figure 4).

Other minor points to address:

1) Please provide a detailed description of how the ensemble members were assigned to the ground state or to different excited states.

A: In the case of Abl1 and its mutants, we assigned ensemble members to the ground state or to “outside of ground state” by measuring their Activation Loop’s (residues 379 to 395) RMSDs compared to the apo ground state reference (PDB 6XR6). RMSD values above 3.5

A were considered outside of the ground state, corresponding to either intermediate conformations or to the I2 conformation (Figure 3 showcases that the distribution of conformations of ensemble members is mostly contained within the boundaries of the Ground to I2 transition).

The 3.5 Å cut-off was determined by analyzing the distribution of RMSDs using a kernel density estimation. When applying this processing method, two clear modes are observed, each corresponding to either the Ground or I2 states. The right-side boundary for the first and most predominant presumed state sits at around ~3.5Å. Hence, this value was chosen for defining the boundary of the ground state.

We added the figure below and accompanying discussion to the Supporting Information:

Fig S1. Distribution of A-Loop backbone RMSD vs. the ground state reference (PDB 6XR6) for the predicted Abl1 kinase ensemble generated by AF2 with subsampling conditions 512:1024 (top) or 256:512 (bottom). Frequencies are calculated from a kernel density estimation with 480 samples per ensemble (96 independent seeds * 5 different models).

2) How were the “regions of putative high mobility” determined based of the chemical shift perturbation data in Figure 8?

A: We determined the regions of high mobility by visual analysis of the chemical shift perturbation data layered in the structural model of GMCSF (which is now better highlighted in the modified version of Figure S8, added after revision). Specifically, we looked for residue ranges with significant peaks in the CSP plots and/or a high number of peaks broadening.

3) Please plot the chemical shift perturbation data from Figure S7 as a color gradient on the atomic resolution structure. Also, acknowledge that these chemical shift changes (as well as the observed line-broadening) may originate from conformational changes involving the surrounding side chains, and that other experiments (i.e. RDC) should be run to confirm that the chemical shift perturbations are the result of changes in backbone dynamics.

A: We have amended Figure S8 to include the suggested color gradient on the atomic resolution structure. We also added the following passage to the main text to acknowledge that the CSPs might originate from conformational changes involving side chains and how to confirm results.

Although the magnitude and distribution of the $1H$ - $15N$ peaks observed in Figure 8 suggest that significant conformational changes involving backbone atoms might be happening in GMCSF mutants (especially those involving H15 and H83), we must not discard the possibility that the side chains surrounding the measured atoms might be inducing or involved in the conformational changes. To distinguish between contributions from the main and side chains, further studies exploring GMCSF dynamics with methods that show ensemble averaging of backbone structures such as residual dipolar coupling might be necessary.

4) Did the author make any attempt to measure the relative populations of ground and excited state in GMCSF by NMR? These values would make the comparison with the computational predictions much more stringent.

A: We absolutely agree that these values would make the comparison better, and we are working on obtaining this data for a future study that dives deeper into GMCSF dynamics. Since it is difficult to observe alternative conformations that do not amount to significant populations in relaxation experiments, we are moving towards slower timescale measurements such as CEST and/or ensemble studies via residual dipolar coupling to obtain averages of conformations to measure if the observed mutations significantly alter the conformational landscape of GMCSF as expected from the CSP data and from the AF2 predictions.

5) Please discuss best practices to test predictions experimentally.

A: Testing predictions experimentally is system-dependent considering the heterogeneity of the systems that we presumably can explore with subsampled AF2. In the case of GMCSF, slower timescale measurements could help detect significant changes in conformational landscapes that might be mirrored in the AF2 results. Capturing a putative alternative conformation could also be feasible if it has sufficient occupancy in the mutants; otherwise, it could be trapped with engineered mutations and contrasted with the AF2 results. For enzymes such as Abl1, biochemical assays could be used as proxies to measure the accuracy of AF2's predictions of activate/inactive differences, and binding experiments could be used to test results that are presumed (by analyzing the AF2 output) to cause drug resistance.

Reviewer #2 (Remarks to the Author):

However, the study is extremely small (only three examples) so it is not clear to know if this better provide any advantage over earlier subsampling (e.g. by Meiler) or dropout methods (Wallner).

1) The study should be expanded significantly. At the bare minimum all models tested by del Alamo et al should be included. But it could probably be expanded as well. Note that this will not be a reproduction of earlier results, as another AF version was used in their studies.

A: While we agree with the reviewer that our study is similar to previous works that used AlphaFold to sample different conformations, we reiterate the novelty of our observation that subsampled AlphaFold can be used, in some cases, to qualitatively predict the relative state populations of proteins and how these populations change in response to point mutations and evolution. At the time that our study was submitted for publication, no previous method had attempted to predict relative state populations and their changes in response to point mutations without employing additional methods. After revision, we also added comparisons between ensembles of predictions stemming from MSAs with different depths and compositions, and measured how each of the five models that ship with AF2 fares at the relative state population prediction challenge, which are to the best of our knowledge, entirely novel experiments.

We appreciate the reviewer's request and, as described above, have thoroughly examined and presented data for eight more systems in the Supporting Information. However, we believe that "extremely small" is not a fair assessment of our study. While the reviewer is correct that the study in its original form is comprised of three examples (Abl1's orthologs, Abl1's activating and inactivating mutants, and GMCSF destabilizing mutants), each of our tests is multi-pronged due to the comparisons with mutants (eight mutants for Abl1 or GMCSF) or orthologs (Src/Anc-AS), each test of which is composed of three to nine measurements in triplicates. In contrast, the studies cited by the reviewer limit their analysis to the wild-type form of each predicted system.

As each prediction ensemble has a sample size of 480, the number of individual predictions described in the original text of the paper is greater than 10,000, and that is without considering the parameter optimization steps. That being said, we appreciate the reviewer's suggestion of further examples to improve the credibility of our method, and have extended the scope of the study significantly by adding eight new test systems, two of which have accompanying mutants. These systems are described below:

1. Proteins suggested by Reviewer 1: Calmodulin, AlkB, and Fyn-SH3
2. Proteins suggested by Reviewer 2: LAT1, LmrP, and CCR5 (three samples from the del Alamo study), and Carbonic Anhydrase VI (negative control, a knotted enzyme with very slight intrinsic dynamics)
3. Protein identified by us as interesting test case: Aurora Kinase A

Considering the feedback from the other reviewers and the length limitations of the manuscript, we opted to test three models tested by del Alamo and collaborators. Our choices were not random. Firstly, we chose LAT1 for which del Alamo and collaborators

succeeded in sampling both predominant states and in-between conformations, which makes it an excellent control for our method. Secondly, and in contrast, we chose CCR5 because predictions with the del Alamo and collaborators workflow failed to generate structures diverging significantly from the ground state, also presenting an excellent control for contrasting the potential differences between our approaches. Finally, LmrP was chosen due to the fact that, in previous works, the predominant conformation in the predicted ensemble is not the experimentally-known ground conformation, so we opted to test if we could predict the right frequency of the proper ground state conformation.

Other models in the new example set were suggested by reviewers, or we identified as potentially interesting. All results, including statistics for contrasts between wild-type vs. mutant predictions, are summarized in Appendix S2. In summary, we successfully predicted previously-known alternative states and intermediate conformations for each of the systems tested with the exception of Carbonic Anhydrase VI, the negative control. We also successfully predicted relative state populations that correlated with experimental values in a few subsampling conditions for most predictions in the new test set, even when previous methods were not able to do so (for example, in the LmrP test). Finally, we also successfully predicted the effects of mutations in the relative state populations of Fyn-SH3 and Calmodulin, and the differences in conformational state preferences between Aurora Kinase A and Abl1.

2) A negative set (i.e. proteins with a single conformation) is absolutely necessary to include. It is not obvious how to test this, but I would argue that superfamilies with many known protein structures and very little structural variation could form such a negative set.

A: We agree with the reviewer as to the necessity of a negative test, and have included results for our subsampling approach applied to predicting the conformation of the Carbonic Anhydrase VI enzyme, which is highly-ordered, knotted, and predominantly occupies a single conformation [4]. The results of these predictions are showcased in Figure S27. Notably, our heuristic predicted a single, dominant conformation for CA in the vast majority of subsampling conditions. Significantly unfolded and likely unphysical CA conformations were predicted at extreme subsampling levels (max_seq:extra_seq of 8:16 or lower), and importantly did not lead to clearly-defined distinct states as observed in other examples.

As further evidence of the accuracy of our approach, we note that our subsampled AF2 approach is predicting conformational changes in regions of proteins that are known to undergo conformational changes, while maintaining the rigidity of structural elements known to be rigid in most of our examples. This is evidence that our predictions are not random but presumably guided by the coevolutionary signal. This is illustrated by the newly-added Figure S28 that shows that residue ranges 419 to 439 for the Abl1 kinase core and 150 to 200 for AlkB are mostly rigid across our prediction ensembles, regardless of changes in other parts of the structures.

3) Statistics is needed, i.e. a comparison of variation for proteins assumed to have multiple conformations vs. others. How is this biased using different subsampling techniques (values) (as in Fig 1A but on a larger scale), with and without dropouts

etc.. It would also be good to include other versions of AlphaFold as they behave differently.

A: We have included results of the Kruskal-Wallis H-test to measure the degree of difference between the distributions of observables of the wild-type and mutant prediction sets (where applicable) for the Abl1, GMCSF, Fyn-SH3 and Calmodulin cases. The results are described in Table S5.

This statistical test was chosen because it is unlikely that RMSD values will follow a normal distribution considering the nature of energy landscapes (which are more likely to exhibit multimodal distributions). We also show how changing the subsampling conditions affect the conformational distribution for all of the new test cases (Figures S16-S27). We also show how MSA depth and composition affect the Abl1, CCR5, and LmrP predictions (Figures S18-S19).

Finally, we evaluated how results change based on the AlphaFold2 model used, as AlphaFold2 ships with five models (benchmarked for the CASP14 challenge) with significant differences between training regimes. The results of this comparison are outlined in Figure S15, and show that models 3-5 perform better at predicting alternative conformations across a conformational transition. Although it is not within the scope of this study to explore the mechanism behind the better performance of models 3-5, we speculate that it has to do with the fact that these three models were refined without the usage of templates, which reduced the bias introduced by the conformations included in the PDB.

Reviewer #3 (Remarks to the Author):

Predicting Relative Populations of Protein Conformations without a Physics Engine Using AlphaFold 2

The authors showed how to use AlphaFold 2 to directly predict the relative populations of the two proteins under consideration. They tested the approach on (1) Abl1 kinase and (2) granulocyte-macrophage colony-stimulating factor and predicted their conformations.

[1] The abstract is a bit too positive given that it makes no mention of the issues that the method encountered, such as the inability to create samples within the Abl1 inactive state 2 (I2), mis-ranking of some mutants, non-quantitative interpretability of the ranking of the mutant, and similar issues with granulocyte-macrophage colony-stimulating factor (GMCSF). Understandably, all of these cannot be put in the abstract, but some of the issues should. It is left to the authors to decide which ones to include.

A: We agree with the reviewer that, in hindsight, the abstract reads as too optimistic. We have changed it to address the caveats and limitations we have identified and discussed in the study.

This paper presents a novel approach for predicting the relative populations of protein conformations using AlphaFold 2, an AI-powered method that has revolutionized biology by enabling the accurate prediction of protein structures. While AlphaFold 2 has shown

exceptional accuracy and speed, it is designed to predict proteins' ground state conformations and is limited in its ability to predict conformational landscapes. Here, we demonstrate how AlphaFold 2 can directly predict the relative populations of different protein conformations by subsampling multiple sequence alignments. We tested our method against NMR experiments on two proteins with drastically different amounts of available sequence data, Abl1 kinase and the granulocyte-macrophage colony-stimulating factor, and predicted changes in their relative state populations with more than 80% accuracy. Our subsampling approach worked best when used to qualitatively predict the effects of mutations or evolution on the conformational landscapes and well-populated states of proteins. It thus offers a fast and cost-effective way to predict the relative populations of protein conformations at even single-point mutation resolution, making it a useful tool for pharmacology, NMR analysis, and evolution.

[2] The authors mentioned that they systematically tested the "accuracy of different AF2 parameter combinations for predicting the Abl1 kinase core structural ensemble." Were the same set of parameters used for both protein systems studied? How can one be sure that the method presented in the paper is not overfitted to Abl1 kinase or to the granulocyte-macrophage colony-stimulating factor? How should one prove that the method is generalizable to all proteins?

A: This is an excellent point that was also raised by other reviewers. Importantly, we do not argue that our approach is generalizable to all proteins. That being said, we have included an example set with 8 additional protein systems, described in Appendix 2 of the Supporting Information, showing how our approach fares at predicting conformational distributions for a set of proteins with significant diversity of size, nature of conformational change, evolutionary history, and function.

Our method successfully predicted previously-known alternative conformations for all proteins in this new test set (with the exception of Carbonic Anhydrase VI, the negative control), as well as intermediate conformations in-between these known states for most of the proteins in the test suite. We also successfully predicted the effects of mutations in the relative state populations of Fyn-SH3 and Calmodulin, and the differences in conformational state preferences between Aurora Kinase A and Abl1.

As a caveat, although the results of the new test set were promising, for some examples such as LmrP and LAT1, choosing the proper subsampling parameters for accurate predictions of relative state populations without prior knowledge of the system's dynamics was not as trivial as it was for Abl1 or GMSCF. To remedy this, in the discussion of these systems we have suggested heuristics to guide the choice of parameters in the case of ambiguous results with different subsampling levels.

Ultimately, the fact that we have been able to successfully find parameters and predict different conformations and relative state populations for such an array of proteins attests to the likely extensive generalizability of our method.

[3] The authors mentioned that they compiled an extensive MSA spanning over 600,000 sequences. How did the authors determine that 600,000 is enough? Will the approach work with a much smaller collection of sequences? Given that the authors

also showed the results for GMCSF where they used only about 120 sequences, the authors should comment on the number of sequences for GMCSF affected its results if it did. If it did not, why not? Did the authors experiment with what the results would be if the number of sequences was smoothly varied, say from 600,000 to 30?

A: Initially, we compared the very deep MSA built from jackhmmer (over 600,000 sequences) to the MSA built from mmseqs2 (about 30,000 sequences) for the Abl1 kinase. We found that the MSA built from jackhammer led to better sampling of conformations in the Ground to I2 transition, as evidenced by Figure S18. We did not test MSAs deeper than 600,000 due to computational expense of building MSAs significantly larger than that and using them to make predictions (we needed 100+ GBs of ram to feed the Abl1 jackhammer MSAs to AF2).

The reviewer is right that we neglected to discuss how the much shallower GMCSF MSA affected the results. We have addressed this comment by adding the following to the main text:

Although we were still capable of sampling different conformational states even with GMCSF's shallow MSA, it is worth noting the occasional prediction of partially unfolded conformations with no experimental analogs. The same did not happen with the Abl1 example, where only extreme levels of subsampling (lower than 8:16) led to partially unfolded structures. We posit that this loss of resolution at aggressive subsampling levels could be a consequence of the extremely shallow input MSA.

Finally, we also compared how the resulting ensembles varied when using a few different MSAs (in depth and/or entropy) to make predictions with AF2, which is summarized in Figures S18 and S19. Below is the discussion for that test:

Crucially, the ensemble resulting from the single sequence prediction leads to mostly unfolded structures that are not similar to the known organization of the Abl1 kinase core (or of any kinase core). This suggests that, in the absence of templates, the presence of a coevolutionary signal from the input MSA is essential for accurately predicting kinase core conformations. In line with previous observations for the CCR5 and LmrP examples, the predictions using the mmseqs2 MSA as input led to considerably fewer intermediate conformation predictions for the Abl1 kinase core than those from the jackhmmer MSA. Interestingly, the truncated jackhmmer MSA designed to be the same depth as the mmseqs2 MSA still led to considerably more conformations along the Ground to I2 path in Abl1 kinase core predictions. These results match recent studies that found that MSA depth leads to increased accuracy in AF2 predictions [5], while also recapitulating previous results that found that MSA entropy (that is, the average distance between pairs of sequences) also plays a significant role. Although it is not in the scope of this study to answer why this is the case, we hypothesize that MSAs with lower entropy cause AF2 to more easily distill the coevolutionary signal pertaining to conformations that would otherwise be lost in MSAs with larger distances between sequences.

[4] The authors mentioned using dropouts (and 32 predictions with independent seeds for each) during inference to sample from the uncertainty of the models. The authors should elaborate. Were the dropout configurations identical to the ones initially used to train the model? Readers would want to know. In addition, what would

be the effect of higher dropout probability or lower dropout probability on the distribution of the conformation space of the protein sampled?

A: The dropout configurations were identical to the ones set by Wallner (Bioinformatics, 2023) [6], that is, 10% to 25% depending on the network module (Evoformer vs. structural, respectively). We expect that target-specific optimization of the dropout probability for the Evoformer module could be a viable strategy to increase the accuracy of predictions, considering the heterogeneity of multiple sequence alignments for different prediction targets, in terms of depth and entropy between sequences.

For deep MSAs with very low entropy such as the Abl1 kinase core MSA built with jackhmmer (n sequences > 600,000), increasing dropout rates could lead to more efficient learning of the probability space for pairwise representations belonging to different conformations. Alternatively, for sparse and shallow MSAs such as the one built for GMCSF (n sequences < 120), decreasing dropout rates could reduce the variance/prediction of misfolded/inaccurate structures.

We briefly commented on the dropout configurations chosen in the main text in the Conclusions Section, as shown below:

Dropout rates were the same as those found to improve sampling in other studies (10% for the Evoformer module, and 25% for the structural module) [5]

[5] The authors mentioned that "max_seq:extra_seq ratio of 256:512 leads to the most diverse results in terms of activation loop conformations." Do you really mean ratio or these specific numbers? For example, since 256:512, 100:200, 512:1024, ..., are equivalent, would all of them have worked the same way and produce comparable/identical results?

A: We thank the reviewer for catching this blunder in our wording. These are not actually ratios but rather the values of *max_seq:extra_seq* parameters, whose significance is described in Figure 2. We changed the text to reflect this.

[6] The authors mentioned that "none of the Src predictions were found to be in the I2 state, although the enzyme is known to infrequently occupy this conformation. This suggests a resolution limitation in using AF2 to predict relative state populations: conformations with very low occupancy such as I2 in Src might be missed by the algorithm in its current implementation." Would fine-tuning AF2 with new sequences / additional data help? It may help the readers to comment further on these.

A: Yes. We hypothesize that fine-tuning AF2 models by training with significantly more sequences for a protein domain/structural motif would significantly improve the resolution of relative state population predictions. Importantly, the composition of the structural training set must also be taken into account, as Src is overwhelmingly present in the PDB in the ground conformation, so not accounting for this distribution/diversity in the training set could also lead to biases that would limit AF2's resolution in detecting low-occupancy alternative states.

The discussion below was added to the paragraph quoted by the reviewer:

We anticipate that fine-tuning or retraining AF2 and similar AI methods with significantly more diverse structural datasets representing different conformational states of a given protein domain could be a viable strategy for increasing the resolution and accuracy of predicting the conformational plasticity of that domain. Additionally, using deeper MSAs in the training could also improve prediction accuracy, in line with recent results that used extremely deep MSAs to achieve higher prediction accuracies than earlier methods [6]

[7] Are these all the mutations with known effects on the population of the ground state? If not, why did the authors test these mutations alone? Does the method work for the other mutations (whose effects are known) too?

A: There are more mutations with known effects in the relative state populations of the Abl1 kinase core. We chose these mutations because there's a wealth of data pertaining to their effects on relative state populations and, specifically in the case of the activating/drug-resistance mutations such as E255V+T315I, their importance for human health. We expect that the method might work for other mutations as well, but with the caveat that it failed for M290L, so its accuracy might not be 100% for other mutation sets as well.

[8] In Figure 8A, the authors should add a vertical line to the 180° rotation arrow. The line should help the readers to know the axis of rotation.

A: We thank the reviewer for this helpful suggestion that improved the clarity of our work, and have amended the figure as suggested.

References:

[1] Del Alamo, D., Sala, D., Mchaourab, H. S. & Meiler, J. Sampling alternative conformational states of transporters and receptors with AlphaFold2. *eLife* 11 (eds Robertson, J. L., Swartz, K. J. & Robertson, J. L.) Publisher: *eLife Sciences Publications*, Ltd, e75751 (Mar. 2022)

[2] Xie, T., Saleh, T., Rossi, P. & Kalodimos, C. G. Conformational states dynamically populated by a kinase determine its function. *Science* 370 (Oct. 2020)

[3] Neudecker, P. et al. Structure of an Intermediate State in Protein Folding and Aggregation. *Science* 336, 362–366 (Apr. 2012)

[4] Pilka, E. S., Kochan, G., Oppermann, U. & Yue, W. W. Crystal structure of the secretory isozyme of mammalian carbonic anhydrases CA VI: Implications for biological assembly and inhibitor development. *Biochemical and Biophysical Research Communications* 419, 485–489 (Mar. 2012)

[5] Lee, S. et al. Petascale homology search for structure prediction, *BioRxiv* doi: <https://doi.org/10.1101/2023.07.10.548308> (July 2023)

[6] Björn Wallner, AFsample: improving multimer prediction with AlphaFold using massive sampling, *Bioinformatics*, Volume 39, Issue 9, (September 2023)

Reviewer #1 (Remarks to the Author):

The rebuttal letter addresses my concerns appropriately. However, most of the points I raised were only discussed in the rebuttal letter and not appropriately discussed in the manuscript. The authors should implement rebuttal to the following points in the manuscript or SI.

- 1) how populated the high-energy states need to be in order to be detected.
- 2) how well the method performs on larger conformational transitions.
- 3) How were the "regions of putative high mobility" determined based of the chemical shift perturbation data in Figure 8?
- 4) Did the author make any attempt to measure the relative populations of ground and excited state in GMCSF by NMR?
- 5) Please discuss best practices to test predictions experimentally.

Reviewer #2 (Remarks to the Author):

Although I still think it would be of significant value to systematically compare this approach with earlier approaches, I can understand that the computational cost is too high. This paper still adds some value over earlier work even if it does not answer all questions and it does not contain anything from stop it from being published.

Reviewer #3 (Remarks to the Author):

The authors have adequately addressed previous comments in the revised manuscript, which is much better now.

REVIEWERS' COMMENTS

Reviewer #1 (Remarks to the Author):

The rebuttal letter addresses my concerns appropriately. However, most of the points I raised were only discussed in the rebuttal letter and not appropriately discussed in the manuscript. The authors should implement rebuttal to the following points in the manuscript or SI.

R: We thank the reviewer for bringing this to our attention and apologize for the oversight. We have adapted the main text of the manuscript to include discussions pertaining to the points previously raised.

1) how populated the high-energy states need to be in order to be detected.

R: We addressed this point in the main manuscript text, in the Conclusions section, as quoted below:

“Importantly, subsampled AF2 performed better at predicting alternative conformations when used on systems whose high-energy states are more frequently populated. For example, the experimentally-resolved (by NMR) relative populations of Abl1’s and Fyn-SH3’s higher-energy states are 6% and 2%, respectively. While our approach successfully predicted wild-type Abl1’s (I2, inactive) high-energy state using multiple subsampling conditions, our approach was only able to predict the high-energy intermediate folding state of mutant Fyn-SH3 within a narrow range of subsampling conditions. Further, the Abl1 ensemble of predictions often included a small but significant (> 5%) population of putative in-between conformations, while the Fyn-SH3 prediction ensemble did not include such structures (Fig S23). These results indicate that there is some minimum threshold for the relative population between 2 and 6% to be able to detect higher energy states using subsampled AF2.”

2) how well the method performs on larger conformational transitions.

R: We addressed this point in the main manuscript text, in the Conclusions section, as quoted below:

“The diverse protein test set also allowed us to evaluate how our modified AlphaFold2 approach fared at predicting conformational changes of different scales, both in terms of the number of atoms involved and in the expected timescale of the change itself. From this analysis, we observed that subsampled AlphaFold2 fared better at predicting large and slow conformational changes, such as the LmrP or LAT1 channel openings that involve the correlated motions of hundreds of backbone atoms (Figures S17 and S20). For these types of conformational changes, AlphaFold2 predicted a variety of potentially intermediate conformations spanning the transition and both ground and high-energy states for a wide-range of subsampling values. For faster or less significant conformational changes, such as the conversion between the Fyn-SH3 (Figure S23) intermediate and its folded ground state, subsampled AF2 predicted no potential transition conformations and only predicted both states under narrow subsampling conditions, suggesting a resolution limitation.”

3) How were the “regions of putative high mobility” determined based of the chemical shift perturbation data in Figure 8?

R: We addressed this point in the main manuscript text, in the caption of Figure 8, as quoted below:

“Left: Annotated wild-type GMCSF structure in the ground (closed) conformation as predicted by AF2. Each colored element represents a region of putative high mobility according to chemical shift perturbations (CSPs) in the histidine triad mutants, identified by visual analysis of the CSP data. Specifically, residue ranges with significant perturbations and peak broadening were designated as regions of putative high mobility.”

4) Did the author make any attempt to measure the relative populations of ground and excited state in GMCSF by NMR?

R: We addressed this point in the main manuscript text, in the Discussion of GMCSF results, as quoted below:

“To experimentally test these predictions, we believe that further NMR experiments such as chemical exchange saturation transfer (CEST) or ensemble studies via residual dipolar coupling, could be the best way to confirm if this is a metastable GMCSF conformation with physiological functions. Although certainly promising, these experiments are beyond the scope of this study due to their degree of complexity and cost.”

5) Please discuss best practices to test predictions experimentally.

R: We addressed this point in the main manuscript text, in the Discussion of GMCSF results, as quoted below:

“We believe that further NMR experiments such as chemical exchange saturation transfer (CEST) or ensemble studies via residual dipolar coupling could be the best way to confirm if this is a metastable GMCSF conformation with putative physiological functions. Although certainly promising, these experiments are beyond the scope of this study due to their complexity and cost.”

Reviewer #2 (Remarks to the Author):

Although I still think it would be of significant value to systematically compare this approach with earlier approaches, I can understand that the computational cost is too high. This paper still adds some value over earlier work even if it does not answer all questions and it does not contain anything from stop it from being published.

R: We thank the reviewer for the positive feedback and we agree that a future study performing a systematic comparison among different approaches would be valuable.

Reviewer #3 (Remarks to the Author):

The authors have adequately addressed previous comments in the revised manuscript, which is much better now.

R: We thank the reviewer for the positive feedback.